# NITO: Neural Implicit Fields for Resolution-free and Domain-Adaptable Topology Optimization

**Amin Heyrani Nobari**                                                    *ahnobari@mit.edu*
*Massachusetts Institute of Technology*

**Lyle Regenwetter**                                                       *regenwet@mit.edu*
*Massachusetts Institute of Technology*

**Giorgio Giannone**[*]                                                    *ggiorgio@mit.edu*
*Amazon*
*Massachusetts Institute of Technology*

**Faez Ahmed**                                                            *faez@mit.edu*
*Massachusetts Institute of Technology*

**Reviewed on OpenReview:**

## Abstract

Structural topology optimization plays a crucial role in engineering by determining the optimal material layout within a design space to maximize performance under given constraints. We introduce Neural Implicit Topology Optimization (NITO), a deep learning regression approach to accelerate topology optimization tasks. We demonstrate that, compared to state-of-the-art diffusion models, NITO generates structures that are under 15% as structurally sub-optimal and does so ten times faster. Furthermore, we show that NITO is entirely resolution-free and domain-agnostic, offering a more scalable solution than the current fixed-resolution and domain-specific diffusion models. To achieve this state-of-the-art performance, NITO combines three key innovations. First, we introduce the Boundary Point Order-Invariant MLP (BPOM), which represents loads and supports in a sparse and domain-agnostic manner, allowing NITO to train on variable conditioning, domain shapes, and mesh resolutions. Second, we adopt a neural implicit field representation, which allows NITO to synthesize topologies of any shape or resolution. Finally, we propose an inference-time refinement step using a few steps of gradient-based optimization to enable NITO to achieve results comparable to direct optimization methods. These three innovations empower NITO with a precision and versatility that is currently unparalleled among competing deep learning approaches for topology optimization. Code & Data: https://github.com/ahnobari/NITO_Public

## 1 Introduction

Deep learning methods have established a near-uncontested dominance in a variety of generative problems, such as image and text synthesis (Rombach et al., 2022; Achiam et al., 2023). In engineering-related design synthesis tasks, despite a wealth of promising exploratory work (Regenwetter et al., 2022; Song et al., 2023; Hu et al., 2024; Wu et al., 2021), the performance of such models remains highly contested. Design synthesis is a particularly challenging problem for statistical models (Regenwetter et al., 2023). This is in part due to the nonlinearities between 'feature space' and 'performance space' – very similar designs may have extreme discrepancy in performance. Additionally, design constraints (which may be safety-critical) cause difficult-to-learn gaps in the distributional support of the design space. For these reasons, direct optimization of design parameters has classically been preferred since many optimization algorithms efficiently navigate complex constraints and objectives  (Boyd & Vandenberghe, 2004).

---

[*]Work completed at Massachusetts Institute of Technology

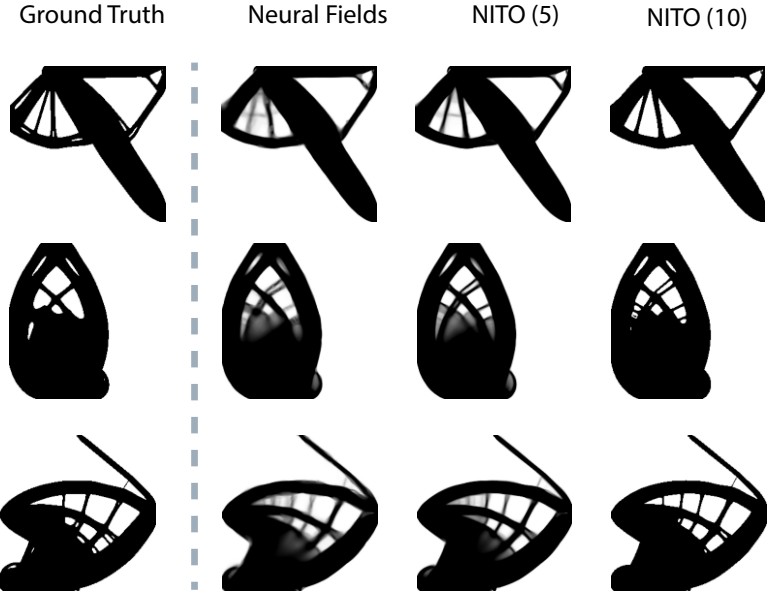

**Figure 1:** NITO framework. Left: Ground truth obtained using SIMP. Second Column: Output of neural fields. Right Columns: the NITO framework output leveraging a few steps (5 and 10 steps) of optimization. NITO rapidly generates high-quality, constraint-satisfying topologies.

The numerous challenges of successful design synthesis using deep learning and the relative success of direct optimization is exemplified by a prominent and frequently-occurring design problem known as structural topology optimization. Structural topology optimization (TO) aims to allocate a finite amount of material within a designated space to maximize structural stiffness, known as compliance minimization. However, depending on the specific engineering application, other performance goals like heat transfer might also be considered. A popular TO method is the Solid Isotropic Material with Penalization method (Bendsøe & Kikuchi, 1988; Rozvany et al., 1992) (SIMP). While methods such as SIMP are powerful engineering tools, they struggle with computational cost when solving large-scale problems due to their iterative nature and expensive per-step computations (Sigmund & Maute, 2013).

In recent years, many works have investigated the prospect of tackling topology optimization problems using machine learning (Regenwetter et al., 2022; Shin et al., 2023). Many of these are methods for direct topology synthesis, which are often trained on a dataset of optimized topologies while conditioned on the corresponding loads and supports that yielded each optimized topology (Nie et al., 2021b; Mazé & Ahmed, 2023; Giannone et al., 2023; Giannone & Ahmed, 2023; Hu et al., 2024). The main justification for these approaches is significant time savings compared to TO solvers like SIMP. This could allow designers to run TO in real time during an interactive design process, remedying frequent pain-points of current direct-optimization TO users (Saadi & Yang, 2023).

Despite early success, deep learning approaches have struggled with several issues. One of these is generalizability, which is critical for their application to real-world TO problems, which are almost always unique. These real TO problems feature one-of-a-kind load and support locations, allow material placement in a specific domain shape and aspect ratio, and require a particular level of detail (i.e., mesh resolution). Most existing deep learning methods discretize the physical domain into pixels or voxels to leverage a convolutional neural network (CNN) as their basic building block (Mazé & Ahmed, 2023; Giannone & Ahmed, 2023; Giannone et al., 2023; Nie et al., 2021b; Hu et al., 2024). This limits their generation capabilities to a specific resolution and shape (such as a square domain). Thus, applying the problem to a new resolution or aspect ratio necessitates an entirely new dataset and model.

In addition to their generalizability issues, pure data-driven approaches typically do not generate optimal topologies compared to classic iterative optimizers like SIMP, despite being faster (Woldseth et al., 2022). This is primarily attributable to the fact that data-driven models typically focus on density estimation and remain agnostic to the physics of a given problem. To address this, researchers often try to incorporate the physics into their model. These approaches typically rely on physical fields (or approximations) as a representation of the loads and supports of the underlying problems (Mazé & Ahmed, 2023; Giannone & Ahmed, 2023; Giannone et al., 2023; Nie et al., 2021b; Chen et al., 2023; Hu et al., 2024). These physical fields are represented as images to be handled by CNNs. Not only is this approach computationally expensive, particularly for high dimensionality, but it further exacerbates the lack of generalizability across problems and domains. These non-generalizable ML-based TO methods are essentially unable to solve new problems and commonly receive criticism from the computational structural engineering community (Woldseth et al., 2022).

In this work, we take a stride toward overcoming key generalizability and optimality shortcomings of ML-based TO. To avoid the generalizability pitfall of CNN-based generative schemes, we propose an approach based on neural implicit fields that can be applied to physical domains of any size, shape, and resolution. We also propose a generalizable scheme for representing sparse loading and support conditions in numerical physics domains, eliminating the need for computationally expensive and domain-limiting physical field calculations, previously thought to be crucial for constraint representation (Chen et al., 2023). Instead, we show that our computationally- and memory-efficient scheme is capable of representing sparse loads and constraints effectively, without loss of performance. To address optimality issues, we propose to couple our deep learning approach with a few steps of direct optimization to refine generated topologies during inference. Our approach outperforms the state-of-the-art in ML-based direct topology synthesis by more than 80% in key performance metrics while running 2.5-50 times faster.

**Contributions:**

1. We introduce a novel framework for TO called "Neural Implicit Topology Optimizer (NITO)," a model capable of generating near-optimal topologies. NITO achieves domain- and resolution-agnosticism through its use of neural implicit fields and our novel "Boundary Point Order-Invariant MLP (BPOM)," It achieves state-of-the-art performance through its use of few-step optimization-based refinement during inference.

2. NITO is resolution-free and domain-agnostic during both training and inference. We show in our experiments that NITO can be trained on multiple resolutions and domain shapes simultaneously and can generate topologies across a variety of resolutions, overcoming a major shortcoming of prior works.

3. We empirically show that NITO is an effective framework, producing topologies with up to 80% lower compliance errors than SOTA learning-based models, while requiring an order of magnitude fewer parameters, allowing 2.5-50x faster inference. These performance metrics are consistent across a variety of resolutions, despite maintaining the same architecture and number of parameters.

4. We implement a new SIMP optimizer for minimum compliance TO. Compared to solvers used to generate existing datasets, our new optimizer produces topologies that are on average 3% lower in compliance, yielding a dataset comprised of significantly more optimal topologies. Our optimizer runs up to 6x faster than widely-used Python implementations. We also introduce a high-quality dataset and improved versions of state-of-the-art models.

## 2 Background

This section will provide background information on topology optimization and neural implicit fields. We introduce previous work using neural fields for optimal topology generation in this section and review existing work using deep learning for TO in the Appendix.

### 2.1 Structural Topology Optimization

In structural applications, TO often aims to minimize compliance (structural deformation) given a set of force loads, supports, and a material volume limit (Liu & Tovar, 2014). These loads, supports, and volume

# Topology Optimization Problem

**Figure 2:** Topology Optimization. Given a domain, loads, supports, and volume fraction, TO aims to find the design variables $\phi$ that maximize performance (such as minimizing compliance $f$), fulfilling all the prescribed constraints and respecting the underlying physics (Static equilibrium). Note that the structure displayed on the left is defined by the design variables which indicate which parts of space have material and what parts of space do not.

limit ('volume fraction') are casually referred to as the problem constraints, which are illustrated in Fig. 2. The goal in TO is to find an optimal structural layout (topology), given the problem constraints. Optimality is calculated as the compliance (deformation per unit force applied) of the structure when subjected to the applied loads and and when the supported locations of the structure are constrained not to move. A prominent technique in TO is the Solid Isotropic Material with Penalization (SIMP) method, introduced by Bendsøe (1989). This method models material properties through a density field, where the density value signifies the amount of material in a specific region. The optimization iteratively simulates the system to evaluate the objective, and then adjusts the density field based on the gradient of the objective score for the field. Mathematically, we represent the density distribution as $\rho(\phi)$, where $\phi$ is a set of design variables, each representing a value within the problem domain $\Omega$. Note that the number of design variables is dictated by the number of elements in $\Omega$ which depends on the underlying discretization, and mesh, in this paper, we use structured meshes in rectangular domains, and refer to resolution instead of element counts. A more in-depth discussion on the topic and comprehensive explanation of the optimization problem are provided in Appendix A. The optimization task is structured as follows:

$$
\begin{aligned}
\min_{\phi} \quad & f = \mathbf{f}^T \mathbf{d} \\
\text{s. t.} \quad & \mathbf{K}(\phi)\mathbf{d} = \mathbf{f} \\
& \sum_{e \in \Omega} \rho^e(\phi)v^e \leq V \\
& \phi_{\min} \leq \phi_i \leq \phi_{\max} \quad \forall i \in \Omega
\end{aligned}
\tag{1}
$$

The objective is to minimize the compliance $\mathbf{f}^T \mathbf{d}$. $\mathbf{f}$ is the load vector and $\mathbf{d}$ the nodal displacement derived from $\mathbf{K}(\phi)\mathbf{d} = \mathbf{f}$, where $\mathbf{K}(\phi)$ is the stiffness matrix contingent on the design variables $\phi$. The constraint $\sum_{e \in \Omega} \rho^e(\phi)v^e \leq V$ ensures the total volume does not exceed a maximum limit $V$ (often expressed as a fraction of the domain volume). The optimization seeks design variables within specified bounds ($\phi_{\min}$ and $\phi_{\max}$) for every element $e$ in the domain $\Omega$. By allowing continuous variation of design variables between 0 and 1, this formulation supports gradient-based optimization. Nonetheless, the expensive finite element simulation (solving $\mathbf{K}(\phi)\mathbf{d} = \mathbf{F}$) at each iteration creates significant computational cost. Notably, solving the linear system of the equations scales cubically ($O(n^3)$) with the number of nodes. This, however, is the cost of solving a fully dense linear system of equations. In reality, linear elasticity results in very sparse symmetric positive definite matrices which are much faster to solve. Specifically, when working with structured meshes like the ones in this work there are many solvers that perform much better than $O(n^3)$ such as the conventional multi-grid approaches proposed by Träff et al. (2023) or even more advanced narrow-band

multi-grid solvers like the one developed by Liu et al. (2018) or more specialized solvers like Aage et al. (2017). In theory, a perfect multi-grid solver could have a convergence rate that yields an $O(n)$ complexity, however, in practice, this is rarely possible. Furthermore, such solvers require algorithms that work iteratively, which makes them not readily parallelizable, hence making the cost of solving each system of equations very high in compute time even if not in complexity. Unlike this, machine learning methods often involve models that are highly parallelizable, and in our case as we will see, the complexity of the proposed methods scales linearly, guaranteeing $O(n)$ complexity while being fully parallelizable. All this does not take into account that a conventional solver has to be employed over potentially hundreds of iterations to solve the optimization problem, while our proposed method can produce structures in one step, and as we will demonstrate can reduce the needed steps for optimization greatly.

## 2.2 Deep Learning For Topology Optimization

The limitations associated with conventional optimization have motivated a significant body of work around deep learning approaches for topology optimization (TO). These approaches have been discussed in detail in dedicated reviews (Woldseth et al., 2022; Shin et al., 2023). Most relevant to our work are other methods that perform the task in an end-to-end manner, meaning they take loads, supports, and volume fraction as input and produce near-optimal topologies to minimize compliance (Oh et al., 2019; Sharpe & Seepersad, 2019; Parrott et al., 2022). For example, Yu et al. (2019) proposes an auto-encoding GAN approach that uses an autoencoder for topology generation and a GAN for super-resolution which is applied simultaneously. Rawat & Shen (2019) and Li et al. (2019) take a similar approach but use GANs for both initial generation and super-resolution. This is while Sharpe & Seepersad (2019), Nie et al. (2021a), and Behzadi & Ilieş (2021) propose directly generating topologies using conditional GANs and Wang et al. (2021c) introduce U-Net-based architectures for TO. More recently, researchers have looked at using implicit neural fields to generate topologies. Specifically, Hu et al. (2024) use implicit neural representations to produce topologies in their approach called IF-TONIR. In their approach, however, the authors use stress and strain fields for their representations of loads and supports which rely on CNNs and somewhat reduce the generalizability of the implicit neural fields' versatility.

**Conditioning on Physics Fields:** Most of the recent state-of-the-art approaches (Mazé & Ahmed, 2023; Nie et al., 2021b; Chen et al., 2023) have adopted a conditioning method in which the loads and supports of TO problems are represented through physical fields such as stress and strain energy. Although these fields are typically simulation-derived, kernel-based approximations have also been proposed to reduce cost (Giannone & Ahmed, 2023). Regardless, this field-based conditioning approach is essential for these models to generate high-performing topologies. Problematically, field-based conditioning has only successfully been handled by CNN models, which prevent the model from being applied to any new problem with different domain shape or resolution than the training data.

**Diffusion-based Topology Optimization:** New methods that utilize diffusion models have recently achieved significant performance improvements over the aforementioned methods (Mazé & Ahmed, 2023; Giannone & Ahmed, 2023; Giannone et al., 2023). Mazé & Ahmed (2023) were the first to do so and demonstrated that diffusion models are significantly better than GANs at optimal topology generation. They also showed that introducing guidance based on a regression model for predicting compliance and a classifier trained to identify floating material significantly improves the performance of such models. Like iterative optimization, however, diffusion models are slow, requiring the models to be run as many as 1000 times to generate a sample. This puts their value into question when the time savings are not notable compared to the optimizer itself. Giannone et al. (2023) propose aligning the diffusion models denoising with the optimizer's intermediate designs as a way to reduce the number of iterations needed for the diffusion model and show that they can indeed reduce the number of sampling steps significantly.

Despite the upsides of the best-performing models, which are diffusion-based, they lack speed and are all based around CNNs, which means that they treat the problem as images. Furthermore, the mechanism they use for conditioning based on loads and supports is field-based, meaning these methods compute some physical or energy field to represent the loads and supports, treat these fields as images, and employ CNN-based architectures to handle them. This method introduces two limitations. These models generate images instead of density fields, restricting them to a specific problem domain and resolution (e.g. a square-shaped domain treated as 64x64 images) and making them unsuitable for different domain shapes and resolutions.

Additionally, the use of image-based fields for loads and supports further narrows their generalizability. This approach fails to capture loads and supports in their raw form, limiting applicability to problems with loads and supports defined for the exact domain they are trained on and excluding problems with irregular loads and supports.

**Shape Optimization And Neural Operators**   Another approach that has been applied in physics-based optimization of designs is shape-optimization. Such models attempt to predict optimal shapes for physics-based objectives such as aerodynamic properties like lift and drag coefficients Viquerat et al. (2021); Durasov et al. (2024); Wei et al.; Durasov et al. (2023); Baque et al. (2018); Heyrani Nobari et al. (2021); Chen & Ahmed (2021); Chen & Fuge (2021). In such approaches, the objective is often to optimize a shape typically defined by continuous functions such as Bézier curves or splines and in some cases discrete meshes defined as graphs. Given the nature of these problems being defined in function spaces, they lend themselves to neural operators Kovachki et al. (2023) which are machine learning models that map between continuous function spaces, which is particularly useful when the shapes being optimized are defined in such spaces. Neural operators are effective in modeling PDEs and shapes. On its surface, such models provide a road map for resolution-free and adaptable models for structural TO problems similar to the problem we target in this paper. Unlike shapes, however, here we intend to optimize the material distribution in a continuum. This means that a simple function space like Bézier curves or other shape functions would limit the space of structures with detailed features. This is a well-known issue with such representations in the structural topology optimization literature Wang et al. (2021a). As such, in conventional structural TO approaches, function spaces are defined in a different manner. The most commonly explored approach that is analogous to shape optimizations is the level-set methods for topology optimization Wang et al. (2021a). In these approaches, a field function is defined and sampled at a specified level. This means that a continuous field is defined by a continuous function and the existence of a material is based on whether the value of a field is above a set value or level, hence the name level-set. Like shape optimization, this allows for sampling at any resolution for any shape. It is often difficult to come up with level-set functions that enable stable and effective continuum topology optimization, and it is well known that despite being continuous and resolution-free, the details obtained by these topologies are limited by the resolution of the underlying solver Wang et al. (2021a). However, neural fields used as the neural operator or level set function has been shown to be effective for structural topology optimization Sanu et al. (2024); Erzmann & Dittmer (2024); Chandrasekhar & Suresh (2021) using conventional optimization loops. In these approaches, however, a direct neural operator in its conventional form is often not practical, rather a neural operator with a level set addition is usually applied Erzmann & Dittmer (2024). As such, we choose to work with neural fields with level set as our primary approach rather than using neural operators, since no clear function space has emerged in the conventional optimization literature. In the section that follows we describe our approach in more detail.

## 2.3   Implicit Neural Fields

A neural field is a field that is partially or fully defined by a neural network (Xie et al., 2022). This neural network takes some form of coordinate representation in space $\mathbf{x} \in \mathbb{R}^n$ as input and outputs a set of values (i.e. a measure of the desired field) $\Phi(\mathbf{x}) \in \mathbb{R}^m$:

$$\tilde{\Phi}(\mathbf{x}) = f_\Theta(\mathbf{x}) \tag{2}$$

where $f_\Theta$ is the neural network function given parameters $\Theta$. These neural fields have proven effective in representing audio (Du et al., 2021), images (Skorokhodov et al., 2021; Sitzmann et al., 2020), videos (Yu et al., 2022), 3d objects (Park et al., 2019), 3D scenes (Mildenhall et al., 2020), and many more applications (see the review by Xie et al. (2022) for a more comprehensive picture). It has also been shown that neural implicit representations can be directly optimized using gradient-based optimizers similar to SIMP to represent the optimal topology for any given problem (Zehnder et al., 2021; Joglekar et al., 2023; Chandrasekhar & Suresh, 2021), which shows that these implicit neural representations are indeed capable of generating density fields that represent *individual* optimal topologies. Compared to existing work, which represents one individual topology per neural field, we propose a conditional implicit neural field approach that generates *different* optimal topologies based on conditions (i.e., volume, loads, supports).

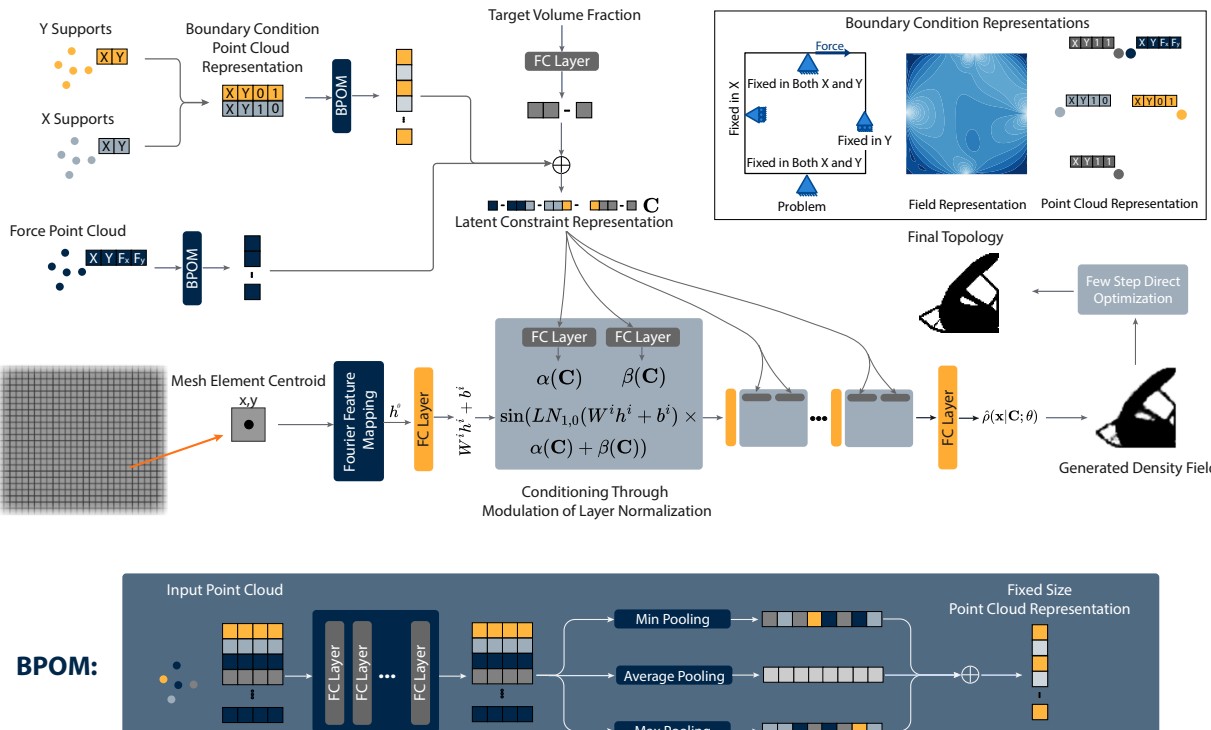

**Figure 3:** The NITO framework for topology optimization: Loads and supports processed as point clouds using BPOM(Illustrated At the Bottom); neural field conditioned on these representations by modulating layer normalization based on latent constraint representation to predict optimal density field; predicted field refined through direct optimization steps. Top-right: Point cloud loads and supports (right) vs. field-based representations like stress fields (middle), given a TO problem (left). Point clouds offer generalizable, memory-efficient representation for loads and supports, unlike expensive, domain-limiting iterative FEA method.

## 3   Methodology

We introduce NITO: Neural Implicit Topology Optimization. Unlike the majority of existing deep learning approaches in TO, NITO effortlessly generalizes across different solution domains and problem resolutions. NITO also achieves a more effective balance of speed and optimality than existing approaches. To enable these achievements, NITO combines three key innovations. First is the neural field-based topology representation that affords much of NITO's flexibility. Second is the Boundary Point Order-invariant MLP (BPOM) which is the second ingredient in NITO's domain- and resolution-agnosticism. NITO's third and final component is a few-step optimization-based refinement of topologies during inference, allowing NITO to rapidly generate solutions that rival the highly-optimal structures generated by direct optimization methods. This section discusses these three key innovations in detail. A detailed schematic of how the NITO framework operates is shown in Figure 3.

### 3.1   Implicit Topology Representation using Neural Fields

The spatial density distribution used in SIMP is highly compatible with neural fields. We denote as $\rho(\mathbf{x})$, where $\mathbf{x}$ is the coordinate of a given point in space and $\rho(\mathbf{x})$ is the density of material in that part of space. Since we need to learn a conditional distribution based on the loads, supports, and volume fraction for a given problem, we model density as:

$$\hat{\rho}(\mathbf{x}|\mathbf{C};\theta) = f_\theta(\mathbf{x}, \mathbf{C}), \tag{3}$$

where $f$ denotes the neural field function which is defined by the neural network architecture and parameters ($\Theta$) and $\mathbf{C}$ is a condition vector that includes information about the domain shape, loads, supports, and desired volume ratio. In practice, this density field is desired to have a density of exactly 0 or 1 at any point,

representing where material should or should not be placed. As such, the density fields that are generated by the SIMP optimizer for a given problem have binary values.

In practice, it is more practical to formulate the problem as the probability of material being placed at any given coordinate. As such, the formulation of the objective can be written as:

$$\mathcal{L}(\theta) = - \mathbb{E}_{\mathbf{x,C}} \left[ \rho(\mathbf{x}|\mathbf{C}) \log f_\theta(\mathbf{x}, \mathbf{C}) \right. \\ \left. + (1 - \rho(\mathbf{x}|\mathbf{C})) \log(1 - f_\theta(\mathbf{x}, \mathbf{C})) \right] \tag{4}$$

where $\rho(\mathbf{x}|\mathbf{C})$ is the probability of material existing at $\mathbf{x}$, which in this case is the same as the output of the SIMP optimizer.

We implement our neural fields using simple multi-layer perceptrons (MLPs) with SIREN (Sitzmann et al., 2020) layers which apply sine activation functions to the output of each layer. Since neural fields tend to ignore higher-frequency features (Tancik et al., 2020), we apply Fourier feature mapping to the input coordinates of the model to mitigate this issue.

### 3.2 Latent Constraint Representation using Boundary Point Order-invariant MLP (BPOM)

Having discussed the neural field implementation, we now consider loads and supports in discretized domains, as seen in TO problems. In particular, we discuss how arbitrary discrete loads and supports can condition generative models in a sparse and generalizable way. This is particularly important for large domains, where defining constraints over grids is computationally cumbersome and memory-expensive.

In most existing studies, authors only considered simple input-only conditioning, which is prone to be ignored by the model (Perez et al., 2017). We suspect this tendency of models to ignore sparse input conditions explains the singular popularity of the rigid field-based conditioning discussed in Sec. 2.2, despite its lack of generalizability. In this paper, we instead propose a simple and effective condition representation model that leverages more advanced conditioning methods to incorporate conditioning information. This allows our model to train on simple and generalizable point-cloud-based conditions, expanding its applicability to various domains and resolutions.

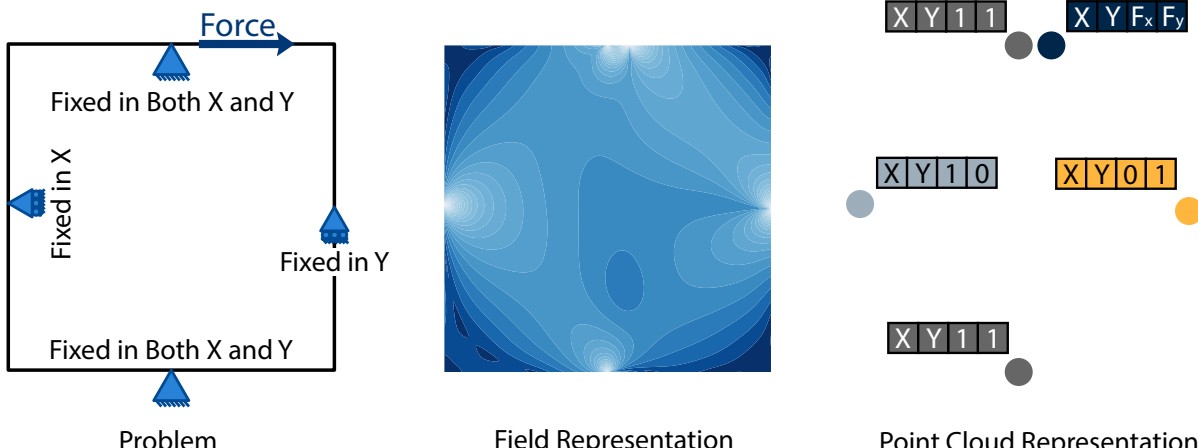

**Figure 4:** Comparison of field-based (middle), and point-cloud-based (right) representations, given a TO problem (left). Unlike the expensive and domain-limiting iterative FEA method, the point clouds offer a generalizable and memory-efficient representation of the loads and supports.

**Loads and Supports as Point Clouds:** In our approach, we represent TO loads and supports as point clouds, which are agnostic to the problem domain and solution resolution. In all problems, conditioning is based on loads, supports, and volume fraction (see Figure 4). Since a problem may have arbitrarily many loads or supports, we need a mechanism that can take all loads and supports for a problem and reduce them to a single latent representation. To do this, we treat these sparse conditions as a set of two point clouds – one for loads, and another for supports (see Figures 3 and 4). Then we process each point cloud using ResP

Layers proposed by Ma et al. (2022) in their point cloud model, PointMLP. However, unlike PointMLP, we removed the geometric affine module (as proposed by the authors for complex point cloud geometries) due to the simpler geometric nature of our sparse load and support point clouds.

**Order-Invariant Aggregation:** Since the point clouds can have any size, we take the output of the point cloud model and perform order-invariant pooling to reduce every point cloud into one single vector representing the loads and supports. We do this by computing minimum, maximum, and average pooling and concatenating the three pooling results. In the end, the three point clouds are each reduced and then concatenated into one vector. We call this representation Boundary Point Order-Invariant MLP (BPOM) and we show that this representation works just as well as physical fields. BPOM's inner workings are illustrated in Figure 3. Finally, for the volume fraction condition (which is always a single floating point value), we use a single fully-connected layer and concatenate its output to the BPOM outputs of the load and support point clouds.

**Feature-wise Linear Modulation:** Next, we discuss how the condition vector is fed into the neural fields model. Neural fields possess the capability to assimilate prior behaviors and exhibit generalization to novel fields through conditioning on a latent variable $\mathbf{C}$, which encapsulates the characteristics of a field. Specifically, this includes the loads, supports and the volume fraction in our problem. Perez et al. (2017) propose Feature-wise Linear Modulation (FiLM) which applies conditioning by modulating the outputs of the different layers of the model. The conditioning mechanism includes two networks $\alpha(\mathbf{C})$ and $\beta(\mathbf{C})$, which predict a multiplicative and additive adjustment to the outputs of each layer. Our implementation is inspired by similar mechanisms built around normalization layers such as adaptive instance normalization (AdaIN) (Huang & Belongie, 2017). Instead of modulating the outputs of each layer, we apply layer normalization and modulate the scale and shift of the layer norm for each feature of the layer output. Our neural field can be described as:

$$f_\theta(\mathbf{x}, \mathbf{C}) = f^{(L)} \circ f^{(L-1)} \circ \cdots \circ f^{(0)}(\mathbf{x}, \mathbf{C})) \tag{5}$$

where $f^{(i)}$ for $i \in \{1, 2, ..., L-1\}$ indicates the function applied at each layer of the neural field except the first and last layer. Each layer in the model takes a hidden state $h^{(i)}$ as input and performs a linear transformation and layer normalization with modulation based on the condition vector:

$$f^i(h^i, \mathbf{C}) = \sin(LN_{1,0}(W^i h^i + b^i) \times \alpha(\mathbf{C}) + \beta(\mathbf{C})), \tag{6}$$

where $LN_{1,0}$ is layer normalization with scale 1 and shift zero and $\alpha$ and $\beta$ are fully-connected (FC) layers that determine the feature-wise scale and shift based on the condition inputs.

BPOM allows NITO to generalize to any domain shape and TO problem as long as that information is contained in the condition latent representation $\mathbf{C}$.

### 3.3 High-quality Solutions using Optimization-Based Refinement

Despite the success of generative models and other deep learning techniques for TO, we observe that the performance of these models always lags behind the optimization baseline. To address this, we follow an optimization-based refinement approach proposed by Giannone et al. (2023) who suggest applying a few steps (5-10) of SIMP optimization to topologies synthesized by deep learning models during inference (compared to 200-500 iterations for full optimization). This allows SIMP to rapidly hone in on more accurate structures for the given loads and supports using near-optimal generated topologies as a starting point. To ensure that the optimization itself is performed optimally, we implement the SIMP method based on the latest research on TO (Wang et al., 2021b) and use this solver for our experiments and training. This optimization-based refinement represents the third and final key innovation of our NITO framework.

### 3.4 Training and Model Details

The training of the conditional neural implicit model is performed in 3 stages. In the first stage, we perform training by sampling points of a 16x16 grid (i.e., we sample points in space that each belong to one of the 16x16 divisions in space for the unit square domain). Furthermore, when sampling, if the material exists in a given sampling grid, we randomly choose a point with material rather than picking points that are void. The first stage of training is carried out for 20 epochs. In the second stage, we sample on a 32x32 grid for 20

epochs. Finally, we train for 10 more epochs, sampling on a 32x32 grid but rather than preferring material points, we sample completely randomly. We use AdamW optimizer with a starting learning rate of $10^{-4}$ which is reduced on a cosine annealing schedule to be reduced at the end of each epoch to reach $5 \times 10^{-6}$ at the final epoch. The learning rate is stepped at the end of each epoch.

In our implementation, we use 8 layers of size 1024 for the neural fields and use 4 layers of size 256 for the three point cloud models (one for applied load and one for supports in each x and y direction). Although intermediate layers of the neural fields model use the aforementioned FiLM activations, the final layer's activation function is a simple sigmoid. Further details can be found in our code.

## 4 Experiments

In this section, we present an assortment of experiments to compare the performance of NITO to existing state-of-the-art models in the literature. With these experiments, we provide compelling evidence that NITO is a scalable, resolution-free and domain-adaptable paradigm that outperforms convolution-based methodologies with a significantly smaller parameter count. We further demonstrate that NITO can generate constraint-satisfying, high-performance topologies much faster than a state-of-the-art SIMP optimizer, while merging the power of deep learning models as an efficient first stage, and the precision and guarantees of optimization as a reliable second stage. For more experiments, discussion, and visualization, we refer the reader to the Appendix.

### 4.1 Experimental Details

Before presenting the results, we will first discuss key experimental details.

**Dataset and Solver:** We create a dataset comprised of various domain shapes and resolutions using a custom SIMP optimizer leveraging recent advancements in iterative TO (Wang et al., 2021b). Noting that much of the previous data-driven TO work uses datasets generated by ToPy (Hunter et al., 2017), which lacks these recent TO advancements and is much slower than our solver, our dataset is larger and more optimal than those used in previous works. We then used this new and faster solver to generate a new dataset with mixed resolution and shapes. To allow for comparison to prior works, we re-created the 64x64 dataset from Mazé & Ahmed (2023) with our solver, finding significant improvements in optimality. We additionally created high-resolution 256x256 data. Finally, we also generated 64x32, 64x48, and 64x16 data, to test NITO's ability to handle mixed domain shapes (something prior works cannot handle). We include loads, supports, volume fractions, and stress and strain energy fields in our dataset. A total of 199,000 samples can be found in our dataset including 48,000 64x64 samples, 61,000 256x256 samples, and 30,000 samples of 16x64, 32x64, and 48x64 each. For each resolution, 1,000 samples are used for testing. To keep results consistent with prior works, we use the exact same 1,000 test samples in the 64x64 subset as prior works have used. For fairness in comparison, we retrained the TopoDiff model (Mazé & Ahmed, 2023) (the best-performing model in literature) on our new re-optimized dataset (64x64 and 256x256 only), while also acknowledging the original performance metrics reported by the authors. Further details on the dataset and solver are included in the Appendix.

**Evaluation Metrics:** We evaluate the models in terms of performance (i.e., minimum compliance) and constraint satisfaction. First, to measure how well the models perform in compliance minimization, we measure compliance error (CE), which is calculated by subtracting the compliance of a generated sample from the compliance of the SIMP-optimized solution for the corresponding problem, then normalizing by the SIMP-optimized compliance value. Since mean compliance error tends to be dominated by just a few samples, we report mean and median compliance error. We also report median volume fraction error (VFE), which quantifies the absolute error between the generated topology's actual volume fraction and the target volume fraction for the given problem. Beyond these base performance metrics, we quantify inference time. The speed of the SIMP optimizer is also benchmarked for comparison. We remove outlying samples, which have extremely high compliance errors (above 1000%) due to models failing to place material in locations where the load is applied. We do this for all models including other SOTA models that we compare against, as is common practice in prior works (Mazé & Ahmed, 2023; Giannone et al., 2023; Nie et al., 2021b).

**Setup:** We train NITO for 50 epochs in four different scenarios: (1) We train on topologies with a resolution of 64x64. (2) We train on 256x256 topologies. (3) We train on both 64x64 and 256x256 datasets simultaneously

to demonstrate resolution-free generalizability. (4) We train on all 64x64, 256x256, 64x48, 64x32, and 64x16 datasets to demonstrate the generalizability of our approach across both resolution and domain shapes. To compare our approach to the state of the art, we also train TopoDiff on the 64x64 in the manner the original authors did (Mazé & Ahmed, 2023). TopoDiff is currently state-of-the-art in literature (Giannone et al., 2023). Training a diffusion model like TopoDiff on large images of size 256x256 is computationally expensive. Despite this, we train TopoDiff on the 256x256 dataset for 500,000 steps and report the results, however, we do not train TopoDiff with guidance (which, as demonstrated in the original work (Mazé & Ahmed, 2023) and corroborated by our experiments, does not lead to any significant improvement in performances).

**Constraint Satisfaction:**  NITO generates fields as its output. In our experiments, we do not explicitly enforce the volume fraction constraint in the problem, rather we simply threshold the field at a value of 0.5 and use that to evaluate the topologies. This is because this aligns with the greedy sampling of the probabilities predicted by NITO. Note that one could enforce the volume fraction constraint on NITO by performing a binary search on the generated density-field to find the threshold that ensures the volume fraction is met. Here we do not do this since this would not be practical to apply to methods that are very close to binary predictions such as TopoDiff and since the threshold search is not strictly part of NITO's training and architecture we don't implement this in our experiments. However, in a practical setting, the cost of binary search is very small and should be incorporated when deploying models such as NITO.

### 4.2 Performance Comparison to State of The Art

**Table 1:** Quantitative evaluation on 64x64 datasets (All models only trained on 64x64 data). All TopoDiff variants are re-trained on optimized topologies obtained using our improved optimizer. *TopologyGAN and cDDPM results from Mazé & Ahmed (2023) and Giannone et al. (2023). w/ G: using a classifier and regression guidance. CE: Compliance Error. VFE: Volume Fraction Error. NITO achieves SOTA performance on topology optimization in terms of compliance and volume fraction error.

| Model | Refinement Steps | CE % Mean | CE % Median | VFE % Mean |
|---|---|---|---|---|
| TopologyGAN* | - | 48.51 | 2.06 | 11.87 |
| cDDPM* | - | 60.79 | 3.15 | 1.72 |
| TopoDiff | - | 3.23 | 0.45 | 1.14 |
| TopoDiff w/ G | - | 2.59 | 0.49 | 1.18 |
| **NITO (ours)** | - | 8.13 | **0.47** | 1.40 |
| **NITO (ours)** | 5 | **0.30** | **0.12** | **0.40** |
| **NITO (ours)** | 10 | **0.17** | **0.071** | **0.25** |

**Table 2:** Quantitative Evaluation on 256x256 datasets (All models are only trained on 256x256 data). The columns are the same as Table 1. NITO is effective at generating topologies with high performance irrespective of the considered problem resolution. On the other hand, the performance of TopoDiff has degraded so much that NITO outperforms it even without a refinement step.

| Model | Refinement Steps | CE % Mean | CE % Median | VFE % Mean |
|---|---|---|---|---|
| TopoDiff | - | 16.62 | 0.59 | 2.92 |
| **NITO (ours)** | - | 9.178 | 0.96 | 1.52 |
| **NITO (ours)** | 5 | **0.25** | **0.09** | **0.34** |
| **NITO (ours)** | 10 | **0.033** | **0.012** | **0.128** |

In Table 1 we present a quantitative evaluation of SOTA models for topology optimization on the commonly-tested 64x64 resolution. Mean compliance error results are on average much higher than median results due

to a small number of samples having extreme compliance. TopologyGAN and cDDPM show higher values in CE % Mean and VFE % Mean compared to other models. TopologyGAN is consistent with prior benchmarks of the model and cDDPM is a naive conditioning of diffusion models, which struggles to perform well in such a complex problem. A vanilla neural fields model without optimization does not perform as well as TopoDiff in mean compliance error, but performs similarly in median compliance error and volume fraction errors. Unambiguously though, Table 1 demonstrates that our NITO framework achieves a significant leap in performance compared to the SOTA. Neural fields start with average compliance errors of more than double TopoDiff. However, after even 5 steps of direct optimization, NITO outperforms TopoDiff and other methods by a large margin. When taking 10 steps of refinement using direct optimization, NITO finds solutions that are on average only 0.1% more compliant than the solutions yielded by 500 steps of optimization.

Next, we proceed to test NITO and TopoDiff on the much more challenging dataset of 256x256 topologies. Table 2 presents the results of this study for NITO and TopoDiff (the leading SOTA approach). In this case, we train TopoDiff with a larger model size to allow for effective learning on the 256x256 images. Despite this, we see a significant performance degradation of TopoDiff with increased resolution. This performance decrease is so severe that the NITO outperforms TopoDiff in most metrics even without a refinement step (highlighted in green in table 2). With refinement, NITO achieves results on par with or slightly better than the 64x64 tests, all while using the same model size, architecture, and training. This illustrates that NITO scales effectively to higher resolutions without performance degradation.

### 4.3 Topology Refinement Across Baselines

**Table 3:** In this table, we consider the effects of few-step direct optimization to topologies generated by TopoDiff. While direct optimization significantly improves the topologies generated by the neural field, it only slightly improves topologies generated by TopoDiff.

| Model | Refinement Steps | CE % Mean | CE % Median | VFE % Mean |
|---|---|---|---|---|
| TopoDiff | - | 3.23 (-) | 0.45 (-) | 1.14 (-) |
| TopoDiff | 5 | 3.55 (+9.91%) | 0.42 (-6.67%) | 0.67 (-41.2%) |
| TopoDiff | 10 | 1.38 (-57.3%) | 0.33 (-26.7%) | 0.45 (-60.5%) |
| TopoDiff w/ G | - | 2.59 (-) | 0.49 (-) | 1.18 (-) |
| TopoDiff w/ G | 5 | 2.24 (-13.5%) | 0.44 (-10.2%) | 0.69 (-41.5%) |
| TopoDiff w/ G | 10 | 1.05 (-59.5%) | 0.32 (-34.7%) | 0.45 (-61.9%) |
| **NITO (ours)** | - | 8.13 (-) | 0.47 (-) | 1.40 (-) |
| **NITO (ours)** | 5 | **0.30 (-96.3%)** | **0.12 (-74.5%)** | **0.40 (-71.4%)** |
| **NITO (ours)** | 10 | **0.17 (-97.9%)** | **0.071(-84.9%)** | **0.25 (-82.1%)** |

NITO uses a few steps of optimization to refine generated topologies. In this section, we consider the performance of SOTA models, should they be subjected to the same few-step refinement using optimization. Table 3 indicates that optimization of topologies generated by TopoDiff yields a much smaller performance boost compared to NITO and even reduces performance in some cases. After just 5 iterations of refinement, NITO outperforms all variants of TopoDiff tested (including TopoDiff with 10 iterations). NITO's lead continues to grow with more refinement steps. Visual examination suggests that while TopoDiff-generated topologies are detailed, topologies generated by the neural field tend to be slightly more blurry (see Fig. 1). Therefore, we hypothesize that Topodiff finds crisp locally-optimal solutions, while NITO, despite being a much smaller model and not using FEA-based physical fields, finds less locally-optimal solutions that lie closer to a stronger 'global' optimum. This suggests that NITO is more robust, generalizable, and efficient.

### 4.4 Inference Speed & Efficiency

Speed is one of the key benefits of ML-based TO over direct optimization. In Table 4 we present a comparative analysis showing that NITO is significantly faster than both iterative TO methods and competing generative models in inference time. For the 64x64 and 256x256 data respectively, resolution, NITO is 83% and 63% faster than the fastest state-of-the-art model DOM (Giannone et al., 2023), and 97% and 96% faster than our

**Table 4:** Average inference time for different problem resolutions. We include 10 SIMP iterations when computing NITO inference time. NITO is resolution-free, i.e. we can leverage the same small model for 64x64, 256x256, and any intermediate resolution. These times are measured using an RTX 4090 GPU and an Intel Core i9-13900K CPU. We run the SIMP optimizer for 300 iterations. For 5000x5000 resolution, a multi-grid GPU SIMP solver is used, and NITO is sampled at 500x500 parallel patches.

| | 64x64 Resolution | | 256x256 Resolution | | 5000x5000 Resolution | |
|---|---|---|---|---|---|---|
| Model | Parameters (M) | Inference (s) | Parameters (M) | Inference (s) | Parameters (M) | Inference (s) |
| TopoDiff | 121 | 1.86 | 553 | 10.81 | - | - |
| TopoDiff w/ G | 239 | 4.79 | 1092 | 22.04 | - | - |
| DOM | 121 | 0.82 | 553 | 7.82 | - | - |
| SIMP (Hunter et al., 2017) | - | 18.12 | - | 316.02 | - | - |
| SIMP (our implementation) | - | 3.45 | - | 69.45 | - | 6,676.06 |
| **NITO (No Ref.) (ours)** | **22** | **0.005** | **22** | **0.16** | **22** | **27.15** |
| **NITO (ours)** | **22** | 0.14 | **22** | | **22** | 245.12 |

fast SIMP implementation (which itself is up to 6x faster than existing implementations). Thus, in addition to generating superior topologies, NITO's speed and efficiency is orders of magnitude faster than the SOTA.

Table 4 also highlights the number of parameters each model uses in their architecture. NITO can be trained on both image resolutions with the same number of parameters, 22 million, achieving SOTA performance, while CNN-based models have to be made larger and still face significant performance degradation. This is further evidence of NITO's efficiency and scalability. We train NITO for both 64x64 and 256x256 resolution for the same number of steps while sampling the same number of points for each batch during training, which means that the model trains roughly for the same amount of time and with the same memory requirements. In fact, a single consumer GPU (we use an RTX 4090) is enough to train NITO. On the contrary, diffusion models like TopoDiff or DOM must grow to match larger resolutions and therefore require more memory and time to run and train. For resolutions above 256x256 or 3D TO problems, these frameworks can be impractical to train or run for most practitioners. In contrast, NITO is built to generalize to different domains/resolutions without issue, allowing for practical training of large problems with consumer-level computational resources. In the following, we specifically showcase the versatility of NITO when handling different domains.

### 4.5 Resolution-Free and Cross-Domain Generalization

| | Training Data | | | | | | | | | | | |
|---|---|---|---|---|---|---|---|---|---|---|---|---|
| | 64x64 | | | 256x256 | | | 64x64 & 256x256 | | | All Data | | |
| Testing Data | CE Mean | CE Med. | VFE Mean | CE Mean | CE Med. | VFE Mean | CE Mean | CE Med. | VFE Mean | CE Mean | CE Med. | VFE Mean |
| 64x64 | 0.17 | 0.071 | 0.25 | 0.27 | 0.11 | 0.30 | 0.22 | 0.072 | 0.29 | 0.22 | 0.074 | 0.31 |
| 256x256 | 0.058 | 0.016 | 0.13 | 0.033 | 0.012 | 0.128 | 0.048 | 0.027 | 0.14 | 0.066 | 0.032 | 0.14 |
| 64x48 | - | - | - | - | - | - | - | - | - | 0.43 | 0.20 | 0.35 |
| 64x32 | - | - | - | - | - | - | - | - | - | 0.90 | 0.20 | 0.41 |
| 64x16 | - | - | - | - | - | - | - | - | - | 1.16 | 0.14 | 0.72 |

**Table 5:** Quantitative evaluation of NITO trained on mixed resolution and aspect ratio data using 10 steps of refinement. **All metrics are reported as percentages**. The results show that NITO trained on mixed-resolution and mixed-shape data performs well across different resolutions and shapes.

To showcase NITO's ability to generalize across multiple resolutions and physical domains, we present results for different training configurations of NITO and its performance on different domains and resolutions. These results are presented in detail in Table 5. We see that the model simultaneously trained on data of all resolutions and aspect ratios performs on par and better than other SOTA models on both the 256x256 data and the 64x64 data while also effectively handling the non-square configurations in the dataset. This illustrates NITO's resolution-free and domain-adaptable nature. It can be trained on multiple domains and

perform well on all of them. Similarly, NITO can train on one resolution and be applied to problems at a new resolution. This capability is demonstrated by testing NITO models trained on 64x64 data with 256x256 resolution tasks, and vice versa. Remarkably, NITO's performance is consistent across resolutions and shapes: it performs similarly when trained on lower-resolution data compared to higher or mixed-resolution data, and likewise when tested on higher-resolution or lower-resolution problems. This underscores NITO's exceptional adaptability, indicating that its architecture not only supports training across multiple domains but also facilitates the transfer of learning from one domain to another, provided the problems share related distributions. This ability signifies a critical advantage of such frameworks—the potential to train on cost-effective low-resolution data and immediately apply or quickly fine-tune the models for higher resolutions.

**Table 6:** In this table, we consider how NITO performs when sampled at much higher resolution of 5000x5000 compared to running SIMP at this native resolution. We see that as expected, NITO's performance degrades in such a scenario, which is expected since the training data is much lower in resolution, however, we see that with a few steps of optimization NITO starts to perform much better, showing how even at higher resolutions sampling NITO can enable faster optimization. **NOTE:** This experiment is conducted on a smaller subset of 10 samples since the cost of running SIMP for such high resolutions is high.

| Model | Refinement Steps | CE % Mean | CE % Median | VFE % Mean |
|---|---|---|---|---|
| NITO w/o Optimization | - | 7.46 (-) | 8.51 (-) | 1.06 (-) |
| NITO | 5 | 2.55 (-65.8%) | 0.56 (-93.42%) | 0.84 (-20.75%) |
| NITO | 10 | 0.78 (-89.54%) | 0.59(-93.07%) | 0.28 (-73.58%) |

Beyond looking at different resolutions and shapes in the training data, we also performed a limited experiment on the use of NITO at resolutions beyond the training data. For this study, we use a limited test set of 10 randomly selected examples from the test data with resolution of 256x256 and run a multi-grid GPU accelerated solver on the same problem at a resolution of 5000x5000 pixels (25M elements). On average, to run the optimizer for 300 iterations, on a RTX 4090 GPU requires over an hour to run (6, 676.06 seconds), which is a rather long time even with GPU acceleration and advanced multi-grid solvers. We then use the NITO trained on all data to sample at this resolution, and perform a few steps of optimization, using the same solver and summarize the quantitative results in Table 6. The quantitative results show that NITO's performance degrades slightly in comparison to the native resolution it was trained on, however, we can still see that warm-starting the optimizer with NITO sampled at a high-resolution does yield results very close to native resolution SIMP optimizer. However, we must make this point that despite the compliance difference being minimal, the level of detail in few-step optimized NITO samples is not like the detail we see in the native SIMP solutions. In Appendix E we visualize an example of this and show this clearly. However, we can see that NITO can be sampled at much higher resolution and much lower cost in comparison to even one iteration of the solver at this resolution (Table 4) which means that given the scalability of NITO this framework serves as a platform for developing models at very high resolutions contingent upon the data being available. Finally, we also present the compute time for each method in Table 4. Note that for 5000x5000 resolution we use a more advanced SIMP solver that runs on GPU instead of CPU and uses a multi-grid solver to accelerate the process. We can see that NITO's inference scales significantly better than the FEA based solver with

## 5    Conclusion & Limitations

In this paper, we introduce Neural Implicit Topology Optimization (NITO), a novel resolution-free and domain-agnostic deep learning framework for topology optimization. Using a neural implicit field representation and our Boundary Point Order-Invariant MLP (BPOM) to represent loads and supports, NITO can be effortlessly trained and deployed on mixed domain shapes and resolutions. Thanks to its few-step optimization-based refinement, NITO significantly outperforms state-of-the-art models in topology optimality across multiple resolutions and domains. Notably, NITO is much faster and has significantly fewer parameters than previous state of the art models. As such, NITO offers a solution to high-dimensional problems that were previously insurmountable with CNN-based methods. Furthermore, NITO's scalability and generalizability offer a robust foundation for future models in topology optimization and other physics-based problems.

**Limitations:** While NITO presents significant advancements in topology optimization, it is not without shortcomings. Firstly, NITO is not a generative model, which means that the outputs of NITO are deterministic and not diverse for a given set of constraints. This limits NITO's ability to generate solutions that enable exploration of the design space. This characteristic could potentially hinder NITO's efficacy in addressing problems outside its training distribution, given the lack of a generative mechanism that enhances performance in entirely new tasks (See the Appendix for a broader discussion). Future work could be devoted to addressing this matter, potentially by leveraging recent advances in generative frameworks for neural implicit fields (You et al., 2023; Kosiorek et al., 2021). In addition, NITO generates topologies that lack sufficient detail before refinement, making it heavily reliant on its refinement step. Future improvements could aim to refine NITO's training procedures and architecture to reduce the reliance on direct optimization. Beyond this, we can see that NITO and prior methods for direct optimization using deep learning models for topology optimization are often trained and work best for lower fidelity structures. Despite our working moving beyond the existing 64x64 limits of prior works, the reality remains that NITO even at a 256x256 resolution, is no where near the very high fidelity optimization of structures seen using conventional methods sometimes extending to billions of elements Liu et al. (2018); Aage et al. (2017). Despite this, the framework of NITO can be sampled at any resolution, and most importantly trained in a scalable fashion for very high resolutions as we discussed before. This means that despite the fact that in this paper we have not explored very high gigapixel/voxel resolutions we have introduced a generalizable and scalable framework which can enable this kind of extension in the future, and we intend to explore few-shot training on very high resolution samples and studying the effectiveness and data requirements for such resolutions.

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

# A   Minimum Compliance Topology Optimization Background

Here, we will briefly discuss the minimum compliance topology optimization problem and clarify some of the terminology and nomenclature used in the main body of the paper. In minimum compliance problems, the objective of the problem is to minimize the deformation under load, i.e., the compliance of a structure while using a specified amount of material. To compute compliance we must first look at the physics behind the problem and how we model compliance.

## A.1   The Physics Model: Linear Elasticity

In this work and most structural topology optimization problems, the physics model we use to model the behavior of a structure under load is the linear elasticity model. In linear elasticity, an assumption is made that a given structure is made up of a material that has the same elastic properties regardless of shape and stress applied to the material. In a simple 1D setup, this is similar to having a spring whose spring constant remains the same regardless of deformation and load applied, and the size of the spring (i.e., Hooke's law). When extending this to structures in a continuum, we arrive at the following partial differential equation (PDE), also known as the linear elasticity (a special case of the Navier-Cauchy equation with constant linear elastic behavior):

$$\frac{E}{2(1+\nu)}\nabla^2\mathbf{u} + \left(\frac{E\nu}{(1+\nu)(1-2\nu)} + \frac{E}{2(1+\nu)}\right)\nabla(\nabla\cdot\mathbf{u}) + \mathbf{f} = \rho\ddot{\mathbf{u}}, \tag{7}$$

Where $\mathbf{u}$ ($\ddot{\mathbf{u}}$ is second derivative with respect to time or acceleration) is the displacement field (a vector field) which determines how much the structure displaces from equilibrium/initial position, $\mathbf{f}$ is a forcing term (also a vector field), which determines the load applied to the martial in the domain of the problem, $E$ refers to the young's modulus of the material a physical constant dictating how stiff a material is, $\nu$ is the Poisson ratio another physical constant dictating how the material deform is directions perpendicular to the load, and $\rho$ is the density of the material. In static or equilibrium analysis, we are interested in the steady-state deformation under load, which allows us to set the right hand side of this equation to zero:

$$\frac{E}{2(1+\nu)}\nabla^2\mathbf{u} + \left(\frac{E\nu}{(1+\nu)(1-2\nu)} + \frac{E}{2(1+\nu)}\right)\nabla(\nabla\cdot\mathbf{u}) + \mathbf{f} = \mathbf{0}. \tag{8}$$

This equation is what we will solve for our problems to determine the displacement under load and compliance.

## A.2   Finite Element: Solving The Linear Elasticity PDE

To determine the compliance of a given structure, we must solve the PDE in Equation 8. This is not possible in closed form, as such, usually this equation is solved numerically. Specifically, the finite-element method is typically applied to solve this equation. In finite element analysis (FEA), we will discretize the domain (often denoted as $\Omega$) of the problem into a mesh, with a finite number of elements and vertices (nodes). FEA then allows us to integrate the PDE in Equation 8 for each element and determine the overall stiffness matrix of a given structure with a right hand side vector dictated by the forcing term $\mathbf{f}$. The details of how this is done in finite elements is outside the scope of this brief discussion, however, the result of FEA is that given material properties $E$ and $\nu$ one can obtain a linear system of equations of the form:

$$\mathbf{K}\mathbf{d} = \mathbf{f} \tag{9}$$

where $\mathbf{d}$ is the displacement vector (can be assembled into the displacement vector field $\mathbf{u}$ in the original PDE) which determines the displacement in each node of a given mesh and $\mathbf{f}$ is the force load applied to the structure. Note that solving the linear elasticity PDE without any boundary conditions is not possible, as such, when performing FEA we will also set the boundary condition of a given problem. In the case of minimum compliance TO the boundary conditions are simply the locations of the supports (i.e., points in space the structure is fixed and not allowed to move). These supports can fix a set of nodes in the mesh and dictate no movement in all or any directions (see Figure 2. Such supports are introduced to the system by adjusting the stiffness matrix in FEA.

### A.3 Optimizing Compliance

So far we discussed how we solve the linear elasticity PDE using FEA, but now we must find a way to optimize the distribution of a finite amount of material within a mesh/domain. In the linear elasticity equation, the stiffness of the material is set to be constant $E$. However, to find an optimal shape within a domain, we allow variation in the Young's modulus based on how much material is placed in an element of the discretized domain (mesh). Specifically, we define the field $\rho$ which determines the density of material in space and adjust the stiffness of each element by setting $E = E_0 + \rho(E - E_0)$, where $E_0$ is a very small stiffness for void/empty elements (needed to prevent singular $\mathbf{K}$ in FEA). By doing this, we are now able to obtain the compliance of a structure by setting the density value for each element in a mesh. It is also important to note that in most cases we do not want continuous values for $\rho$ rather we are interested in binary values determining which elements have material and which ones do not. However, this leads to a combinatorial optimization problem which is not practical to solve in most cases. To overcome this, the Solid Isotropic Material with Penalization (SIMP) method, introduced by Bendsøe (1989) is usually used. In SIMP we model density as a function of design variables $\phi$ which dictate the density with a non-linear exponent penalty $p$, which is usually set to 3. In this setup $\rho^e = (\phi^e)^p$, hence $E^e = E_0 + (\phi^e)^p(E - E_0)$ for element $e$ in the mesh. Based on this and the fact that we want to minimize compliance, $\mathbf{f}^T\mathbf{u}$, the optimization problem becomes:

$$
\begin{aligned}
\min_{\phi} \quad & f = \mathbf{f}^T\mathbf{d} \\
\text{s. t.} \quad & \mathbf{K}(\phi)\mathbf{d} = \mathbf{f} \\
& \sum_{e \in \Omega} \rho^e(\phi)v^e \leq V \\
& \phi_{\min} \leq \phi^e \leq \phi_{\max} \quad \forall e \in \Omega
\end{aligned}
\tag{10}
$$

where $V$ is the target volume/mass of material we desire in the final structure, and $\phi_{min}$ is the lower bound on the design variables, often 0, and $\phi_{max}$ is the upper bound, often 1. This formulation is the common formulation of the minimum compliance problem which we also use in this work. The goal in NITO is to predict $\rho$ directly based on the loads, supports, and target volume fraction.

## B  Additional Experiments

Here, we will discuss some further experiments and discuss some limitations of our approach.

### B.1  Out of Distribution Experiments

**Table 7:** Quantiative Evaluation on out-of-distribution 64x64 datasets. w/ G: using a classifier and/or regression guidance. FS-SIMP: Few-Steps of optimization. CE: Compliance Error. VFE: Volume Fraction Error.

| Model | Train Res. | FS-SIMP | CE % Mean | CE % Median | VFE % Mean | VFE % Median |
|---|---|---|---|---|---|---|
| `out-of-distro` | | | | | | |
| NeuralField | 64 | - | 73.24 | 12.81 | 8.12 | 6.87 |
| TopoDiff | 64 | - | 8.57 | 1.14 | 1.14 | 0.97 |
| TopoDiff w/ G | 64 | - | **7.79** | 1.26 | 1.21 | 1.00 |
| TopoDiff | 64 | 5 | 6.20 | 1.07 | 0.93 | 0.55 |
| TopoDiff w/ G | 64 | 5 | **5.44** | 1.05 | 0.89 | 0.55 |
| **NITO (ours)** | 64 | 5 | 9.33 | 2.37 | 2.22 | 1.32 |
| TopoDiff | 64 | 10 | 2.91 | 0.71 | 0.64 | 0.33 |
| TopoDiff w/ G | 64 | 10 | 2.25 | 0.71 | 0.63 | 0.35 |
| **NITO (ours)** | 64 | 10 | 6.38 | 1.43 | 1.55 | 0.85 |

Something that should be looked at when it comes to these models is their performance generalizability to problems that are very different from the distribution of data used for training. To do this, we test the performance of different models on an out-of-distribution test set for the 64x64 dataset. These results are presented in Table 7 for TopoDiff and NITO. As it can be seen, NITO's performance has significantly deteriorated on out-of-distribution data. This can be attributed to two matters. The first and most impactful is the nature of these models. TopoDiff generalizes better when it comes to these out-of-distribution tests given the fact that the model is generative in nature. This allows TopoDiff to handle out-of-distribution conditions better. To understand this we can look at generative models as a sort of retrieval approach, which allows the models to generate detailed and high-quality samples by finding the most similar topologies from the training data even when faced with very different inputs. On the other hand, NITO is a deterministic model that learns to map specific loads and supports to near-optimal density fields. This makes it rather challenging for models like NITO when it comes to generating topologies for unseen and very different inputs since the mapping is a deterministic one. This highlights the importance of future work focusing on transforming our framework to be generative, which should be possible since many works have shown implicit neural approaches can be made generative (You et al., 2023). The second potential reason contributing to TopoDiff's better performance on out-of-distribution samples is the use of physics-based fields for conditioning. This is because mapping stress fields to topologies makes it easier to handle very different loads and supports since the stress fields may still be similar to samples in the dataset. However, this creates the kind of limitation that we discussed before, as such it is better to focus on making more robust generative schemes rather than using physical fields.

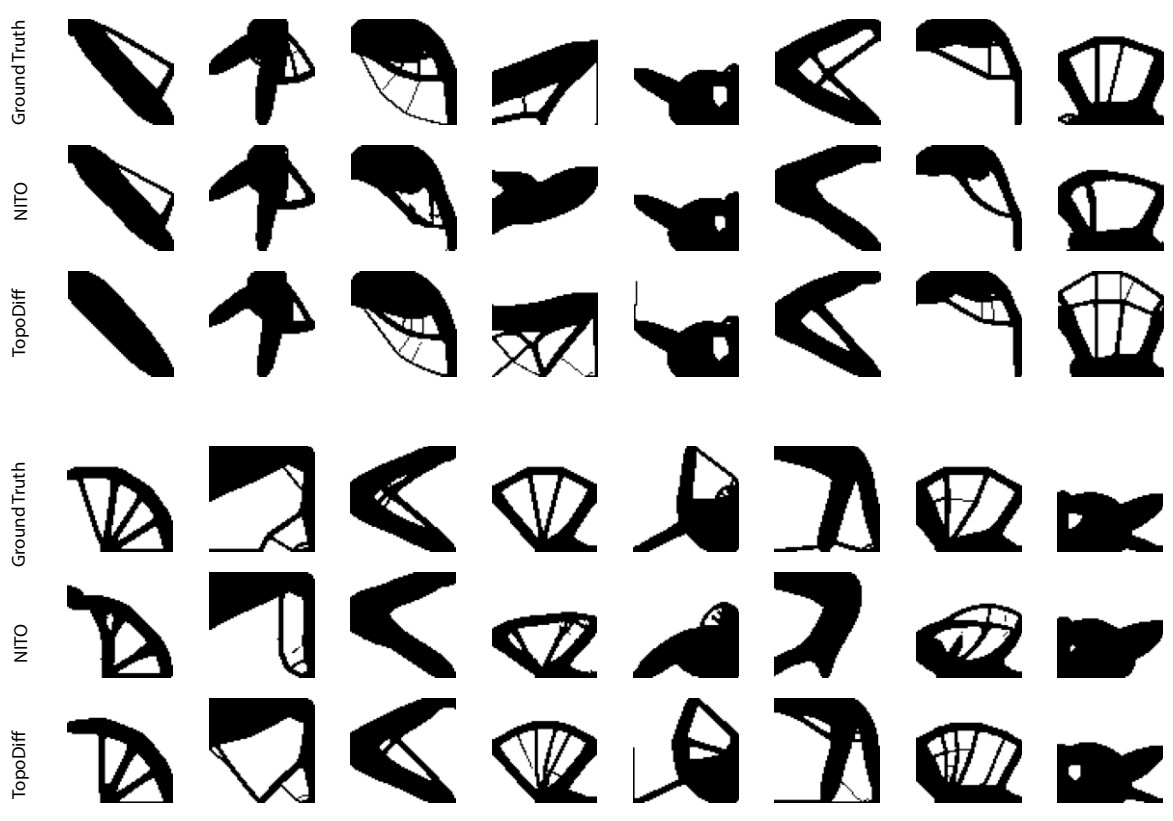

**Figure 5:** Visual comparison of samples generated for the out-of-distribution test. Each row is labeled. Ground truth samples are SIMP-optimized samples.

## B.2  Further Examination Of Results & Outperforming The Optimizer

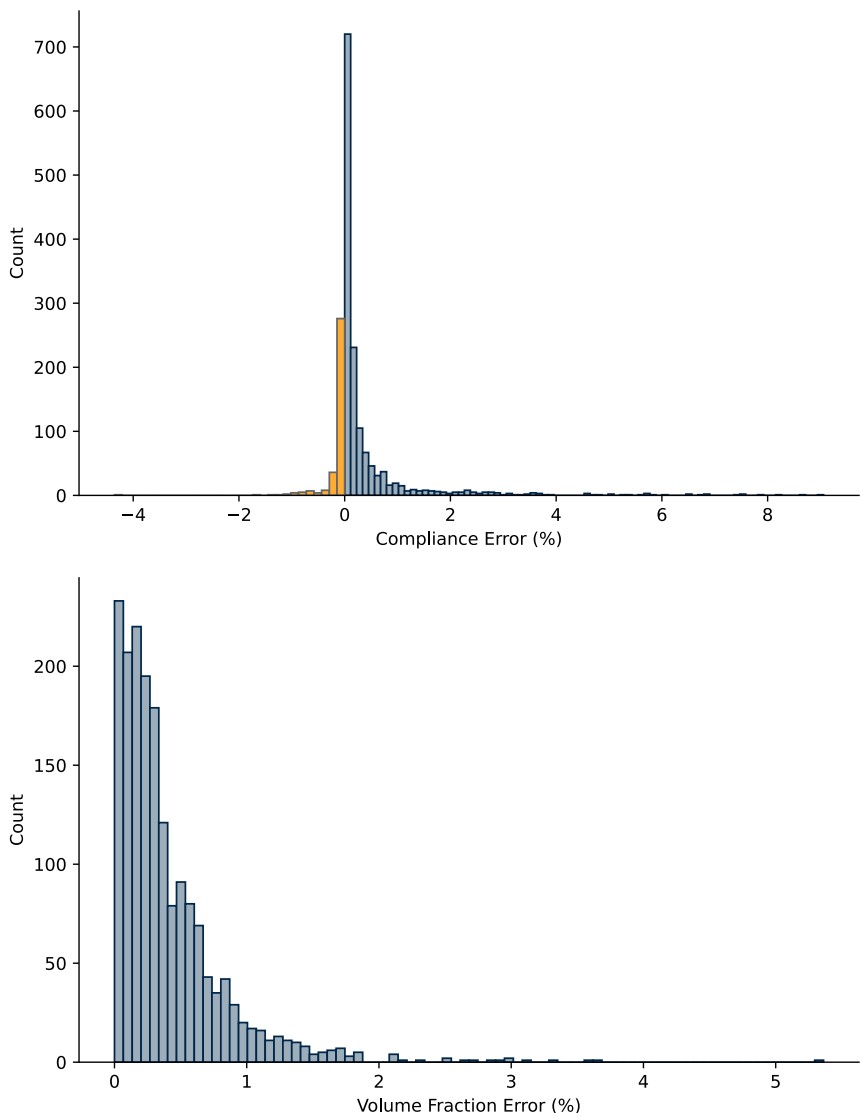

**Figure 6:** Distributions of volume fraction error (Bottom) and Compliance Error(Top). We see that NITO is capable of out-performing SIMP 19% of the time as highlighted by yellow in the negative compliance errors (meaning better than SIMP) on the top histogram. This data is for the 64x64 test set.

In Figure 6, we show the distribution of compliance error and volume fraction error for NITO on the 64x64 dataset. We see that the majority of volume fraction error is below 1% and a small number of high error samples skew the average. Similarly, we see that the majority of the compliance errors for NITO are below 1% as well. Most notably, we observe that on 19.2% of the samples in the test set NITO actually outperforms SIMP as indicated by the negative compliance errors visible on the histogram. This is a rather interesting outcome where NITO is capable of doing better than the optimizer that the training data came from. In Figure 24, we visualize a few instances of this phenomenon. It can be seen that in some problems NITO actually comes up with a different solution which, as it happens, out-performs SIMP, while in other instances the topologies are very similar and NITO has adjusted some of the finer details and redistributed the material differently to achieve better performance.

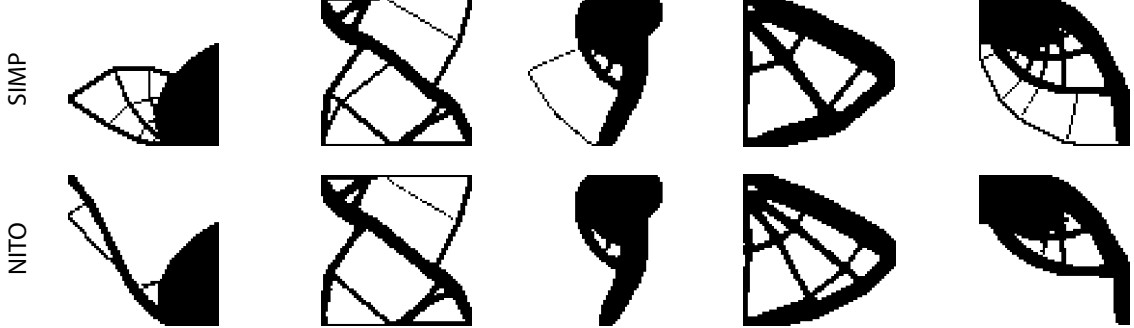

**Figure 7:** Example topologies where NITO outperforms SIMP. It can be seen that in some instances NITO finds very different topologies that outperform SIMP and in some instances, NITO has removed some details and redistributed the material in a way that has improved performance.

# C  Visualizations

Here we will provide a set of different visualizations for each test case.

## C.1  Direct Optimization Visualization

In this section we provide some visualizations, demonstrating the effect of direct optimization on neural field-generated density fields. The main objective here is to showcase the importance and effectiveness of a few steps of direct optimization on the generated samples which completes the NITO framework. As can be seen in Figures 8 and 9, neural fields tend to have some smoothing and averaging in areas of high detail in the topology, which makes the baseline performance of neural fields worse. With only a small number of direct optimization iterations, however, we see that NITO can resolve the complex details effectively, showcasing why direct optimization is a crucial aspect of our framework.

In contrast, when we look at the effect of direct optimization on TopoDiff (Figures 10 and 11) we see that in cases where the details predicted by TopoDiff are accurate, the optimizer does not provide much benefit, and when TopoDiff has not generated the correct details the generated topology is far from optimal which means that few steps of optimization do not provide significant benefits.

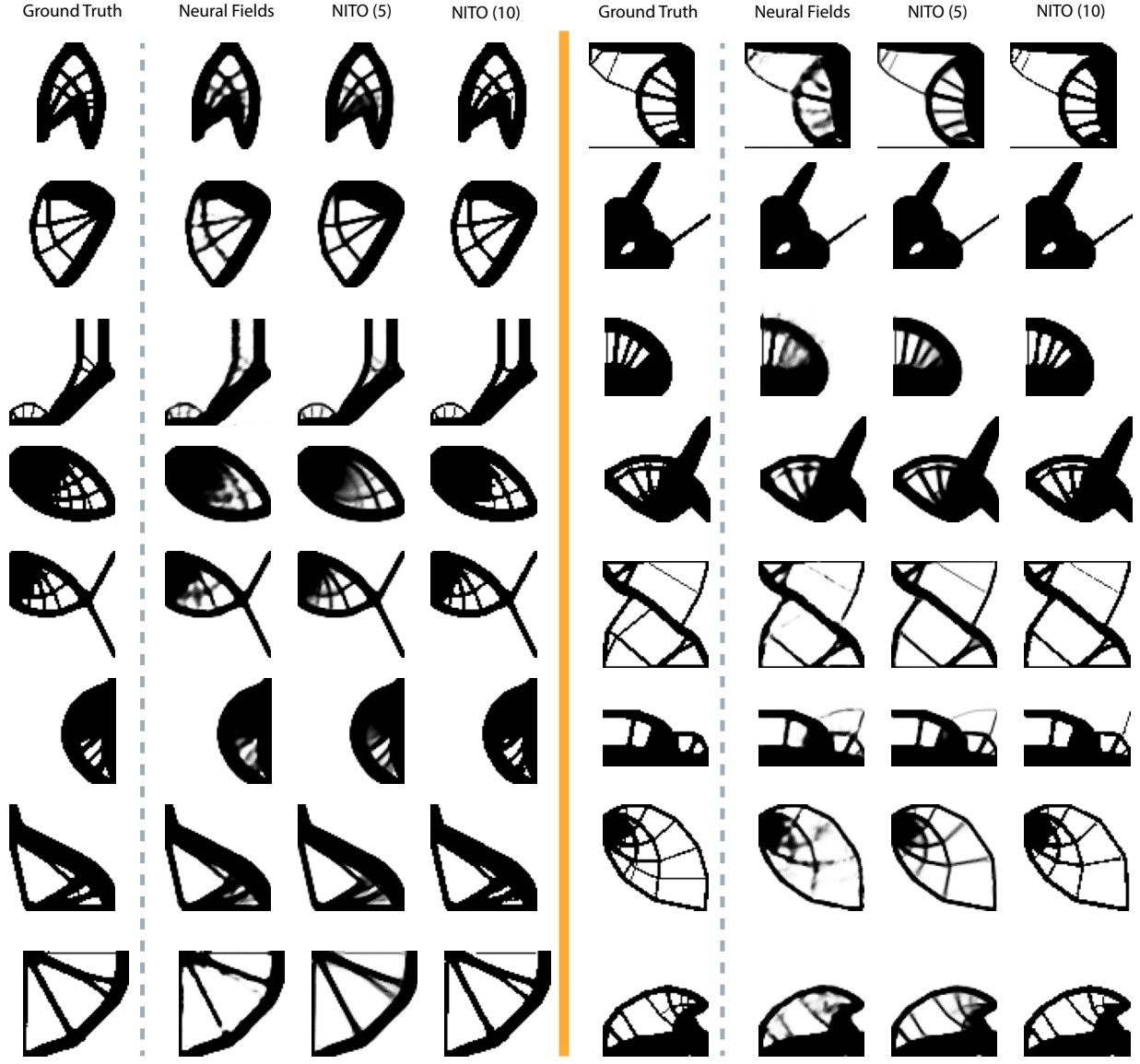

**Figure 8:** Visualization of the effect of direct optimization on samples generated by neural fields. The first column of each set of images shows the ground truth, the next column shows the raw predictions from neural fields, and the two columns that follow show the effects of 5 and 10 steps of direct optimization on the samples respectively. These samples are from the 64x64 test set.

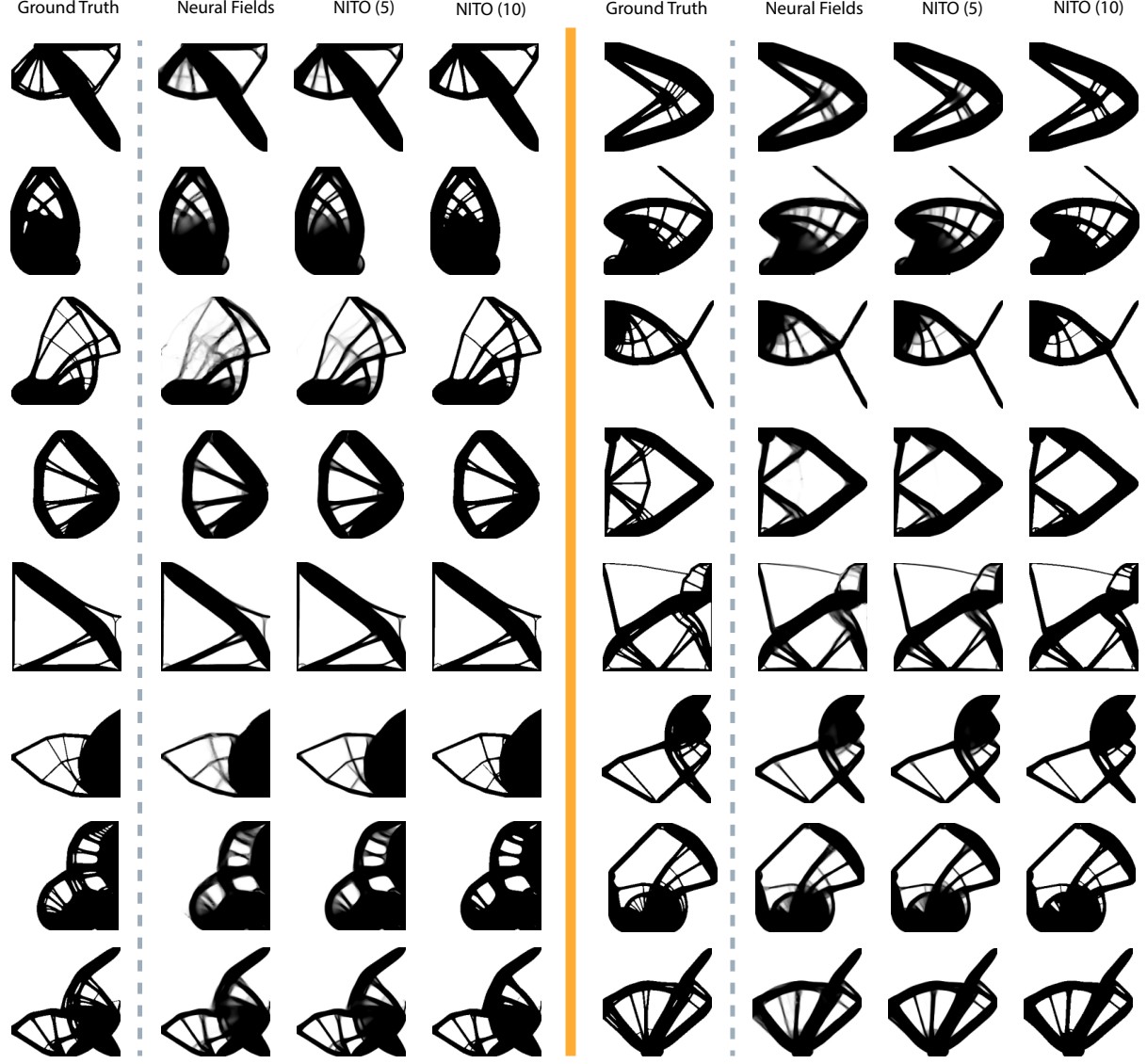

**Figure 9:** Visualization of the effect of direct optimization on samples generated by neural fields. The first column of each set of images shows the ground truth, the next column shows the raw predictions from neural fields, and the two columns that follow show the effects of 5 and 10 steps of direct optimization on the samples respectively. These samples are from the 256x256 test set.

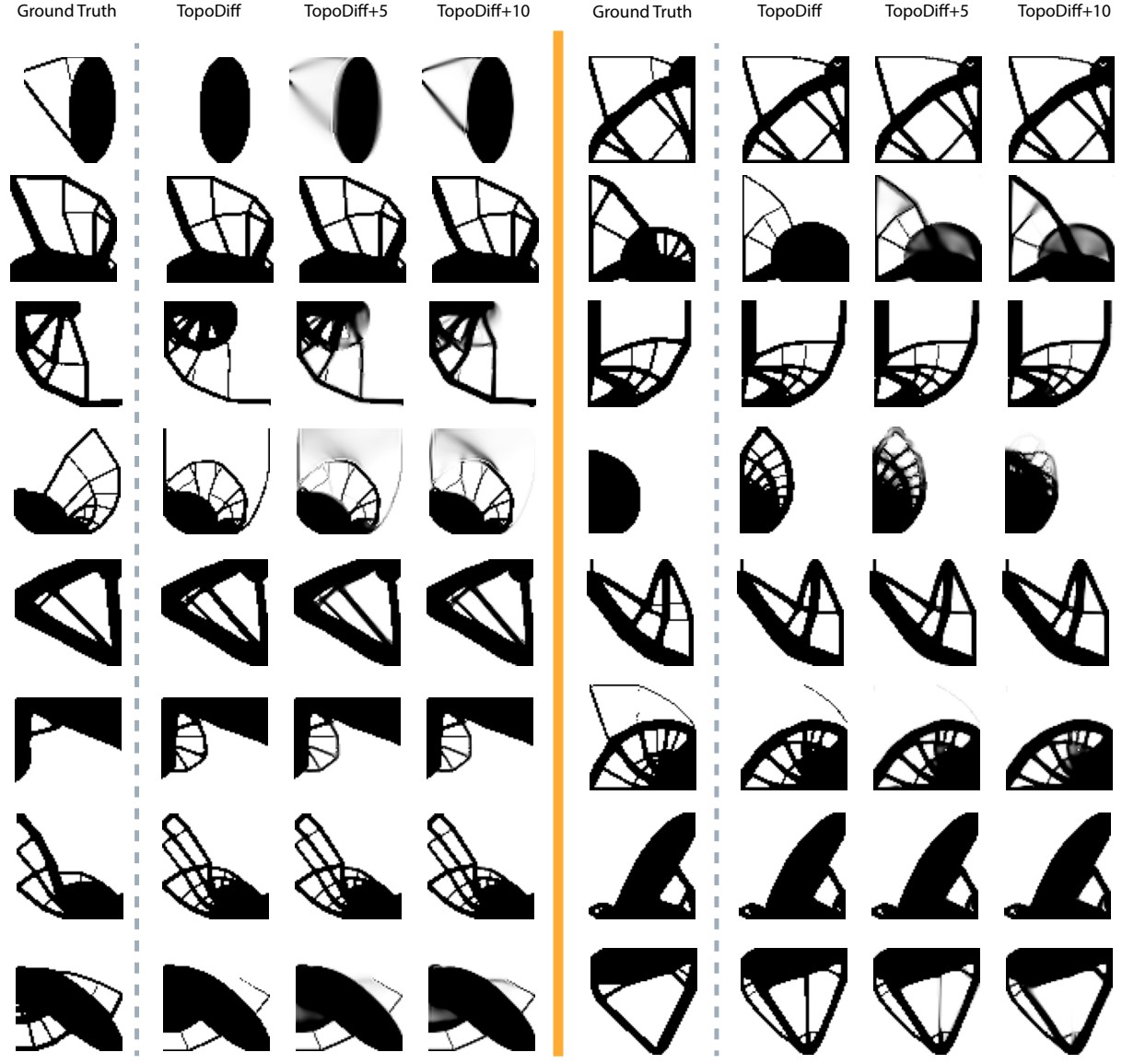

**Figure 10:** Visualization of the effect of direct optimization on samples generated by TopoDiff. The first column of each set of images shows the ground truth, the next column shows the samples generated by TopoDiff, and the two columns that follow show the effects of 5 and 10 steps of direct optimization on the samples respectively. These samples are from the 64x64 test set.

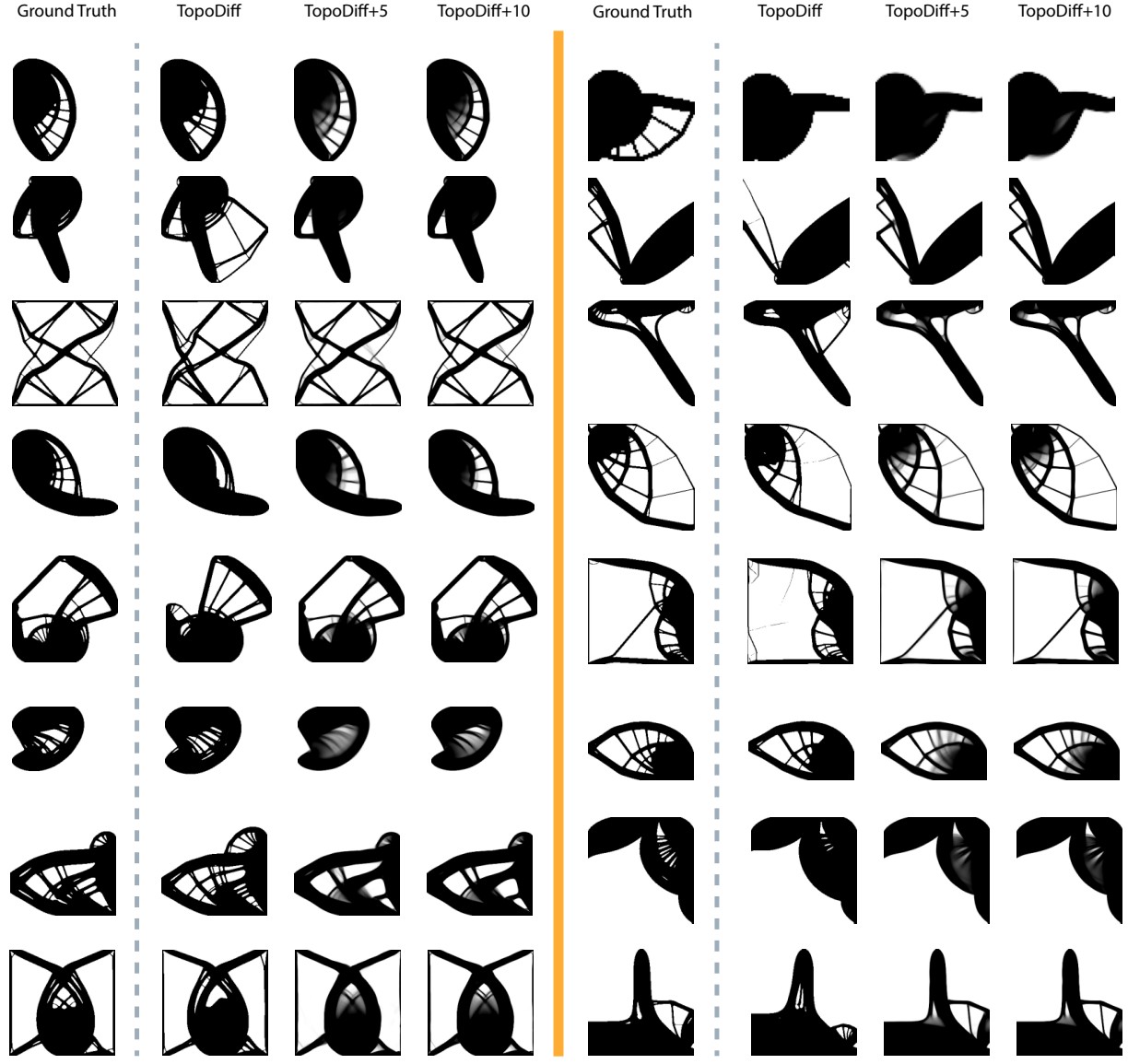

**Figure 11:** Visualization of the effect of direct optimization on samples generated by TopoDiff. The first column of each set of images shows the ground truth, the next column shows the samples generated by TopoDiff, and the two columns that follow show the effects of 5 and 10 steps of direct optimization on the samples respectively. These samples are from the 256x256 test set.

## C.2 Generated Samples Visualizations

In this section, we provide visualizations of samples predicted by different configurations of our model on different problems.

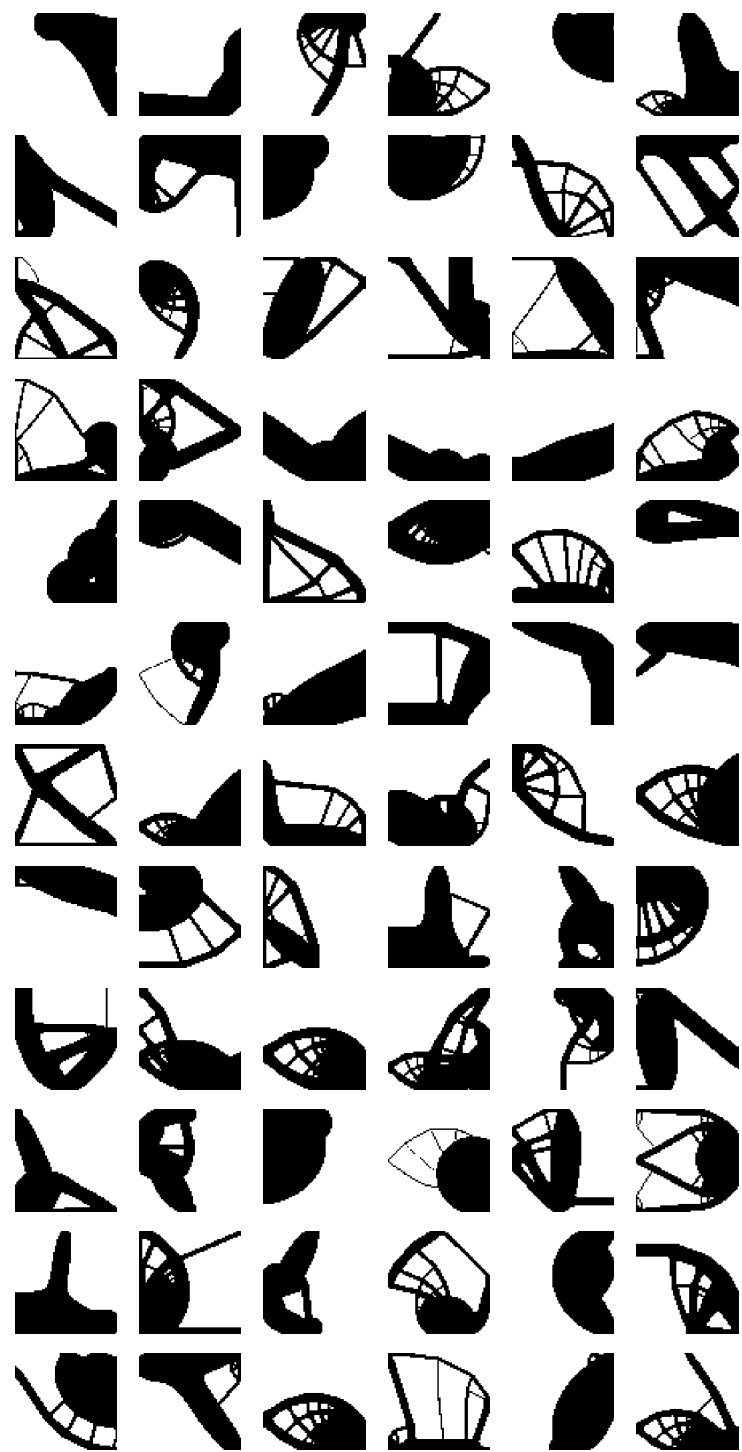

**Figure 12:** Ground truth images from the 64x64 SIMP datasets. Images that follow visualize NITO generated samples for the same problems.

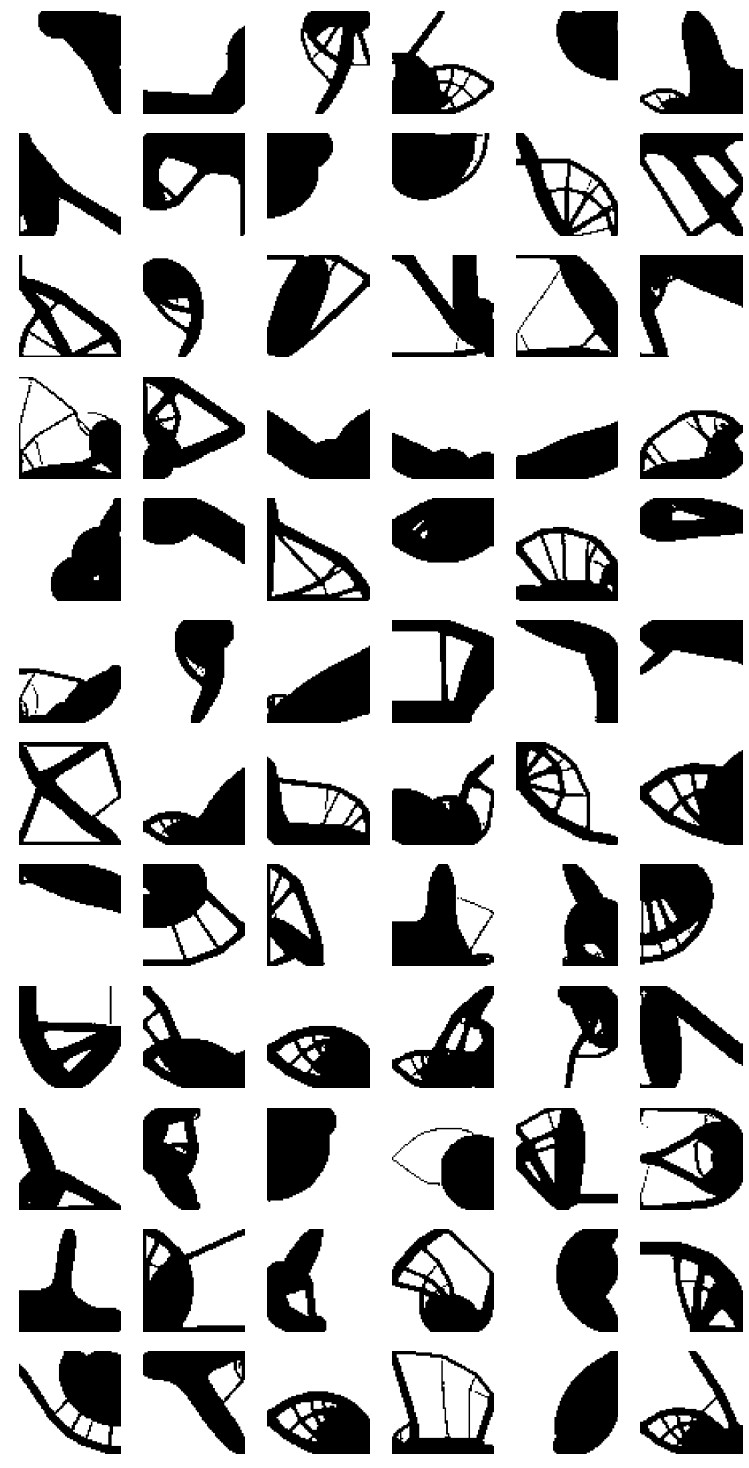

**Figure 13:** NITO generated topologies using a model trained on 64x64. Tested on the 64x64 data.

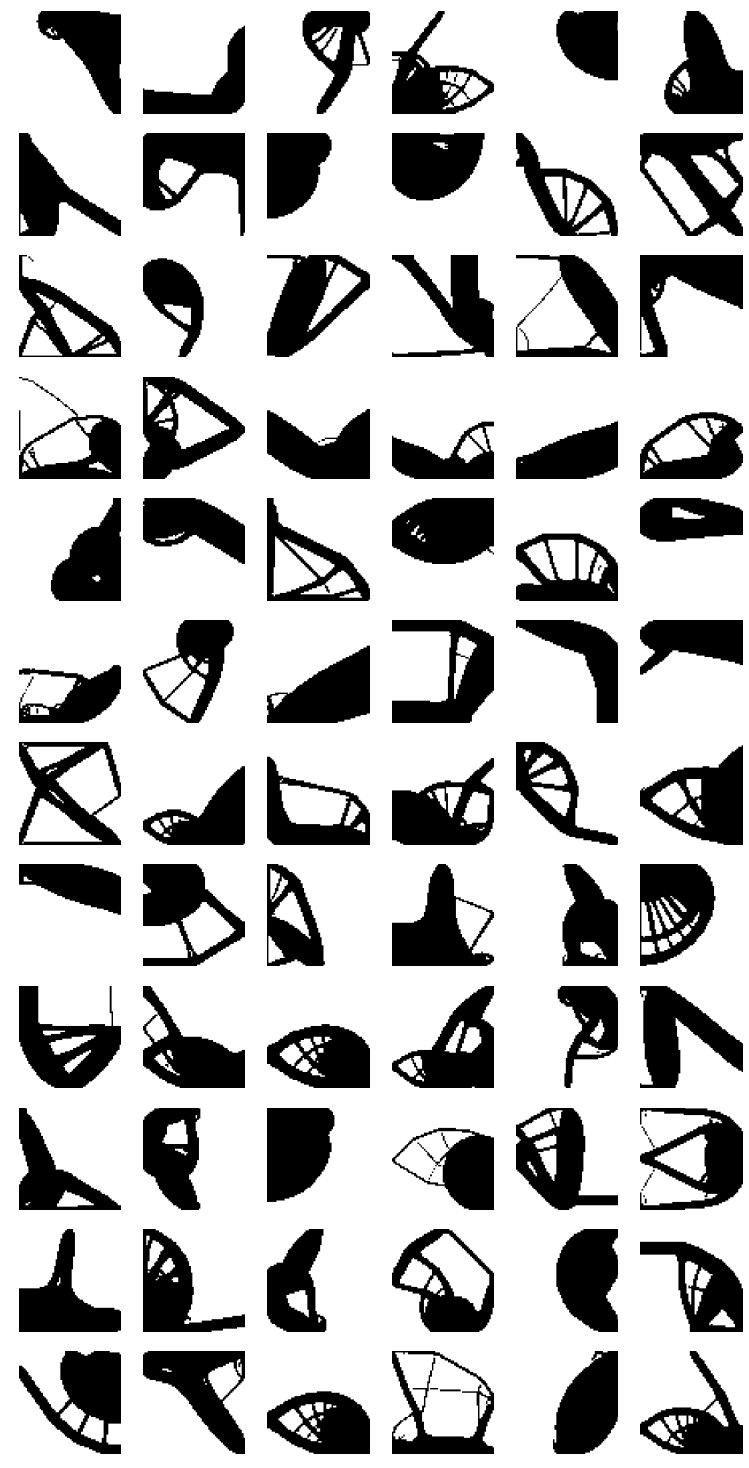

**Figure 14:** NITO generated topologies using a model trained on 256x256. Tested on the 64x64 data.

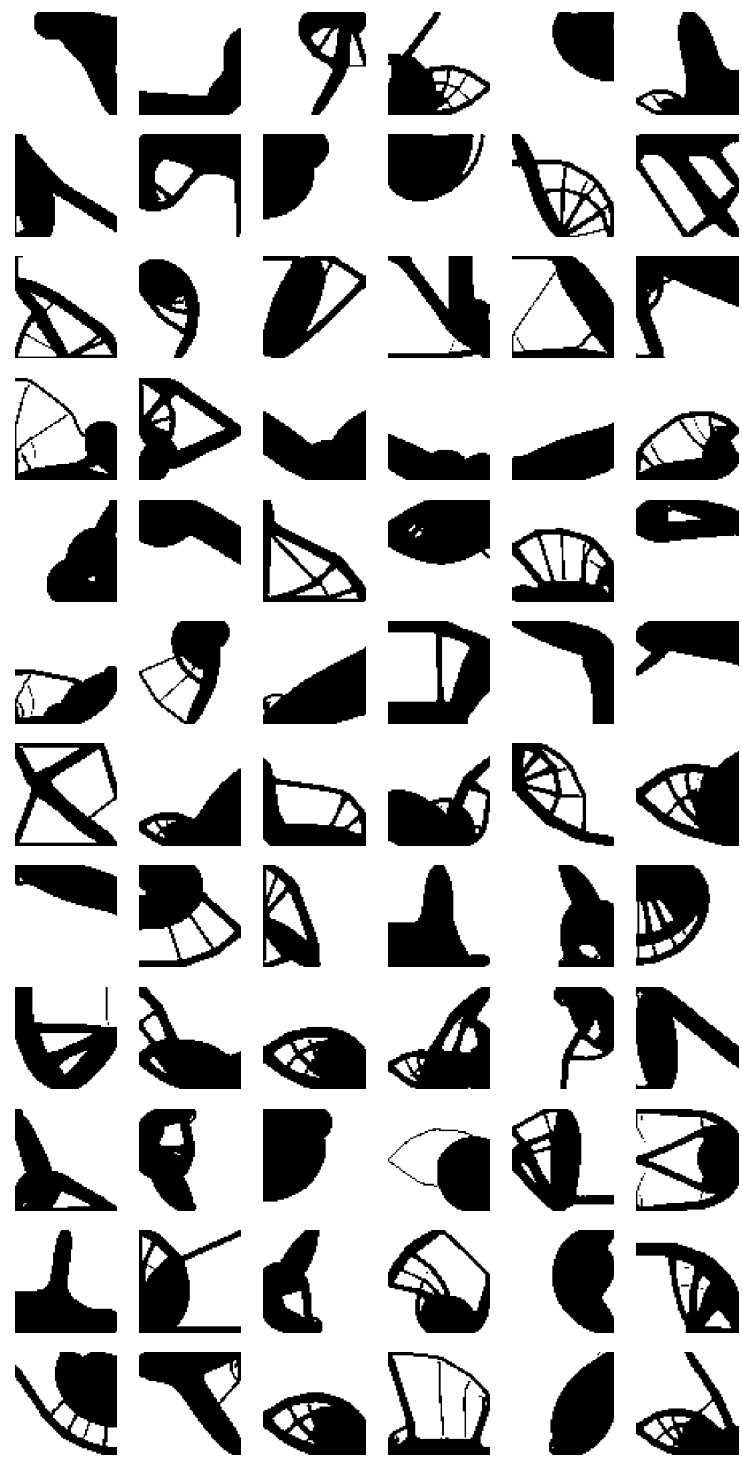

**Figure 15:** NITO generated topologies using a model trained on both 64x64 and 256x256. Tested on the 64x64 data.

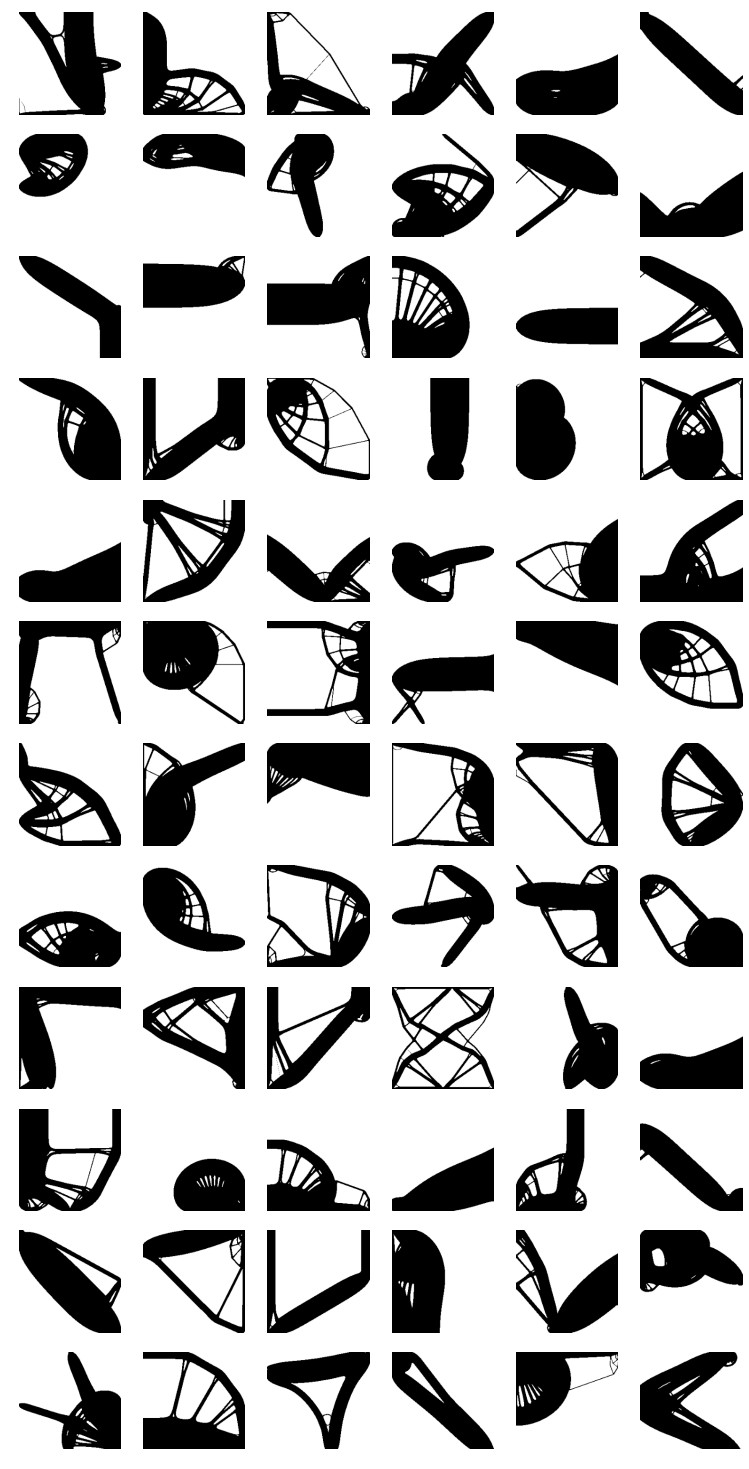

**Figure 16:** Ground truth images from the 256x256 SIMP datasets. Images that follow visualize NITO generated samples for the same problems.

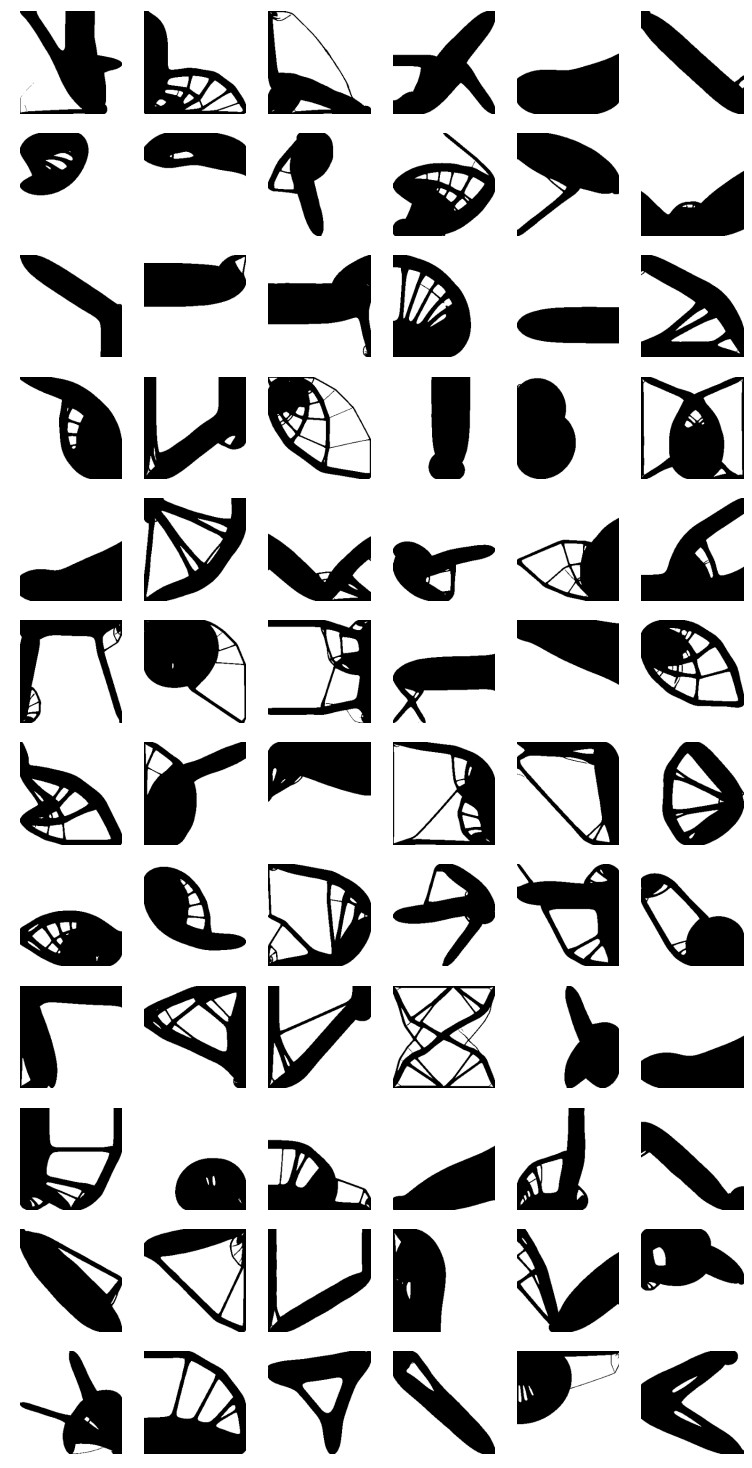

**Figure 17:** NITO generated topologies using a model trained on 256x256. Tested on the 256x256 data.

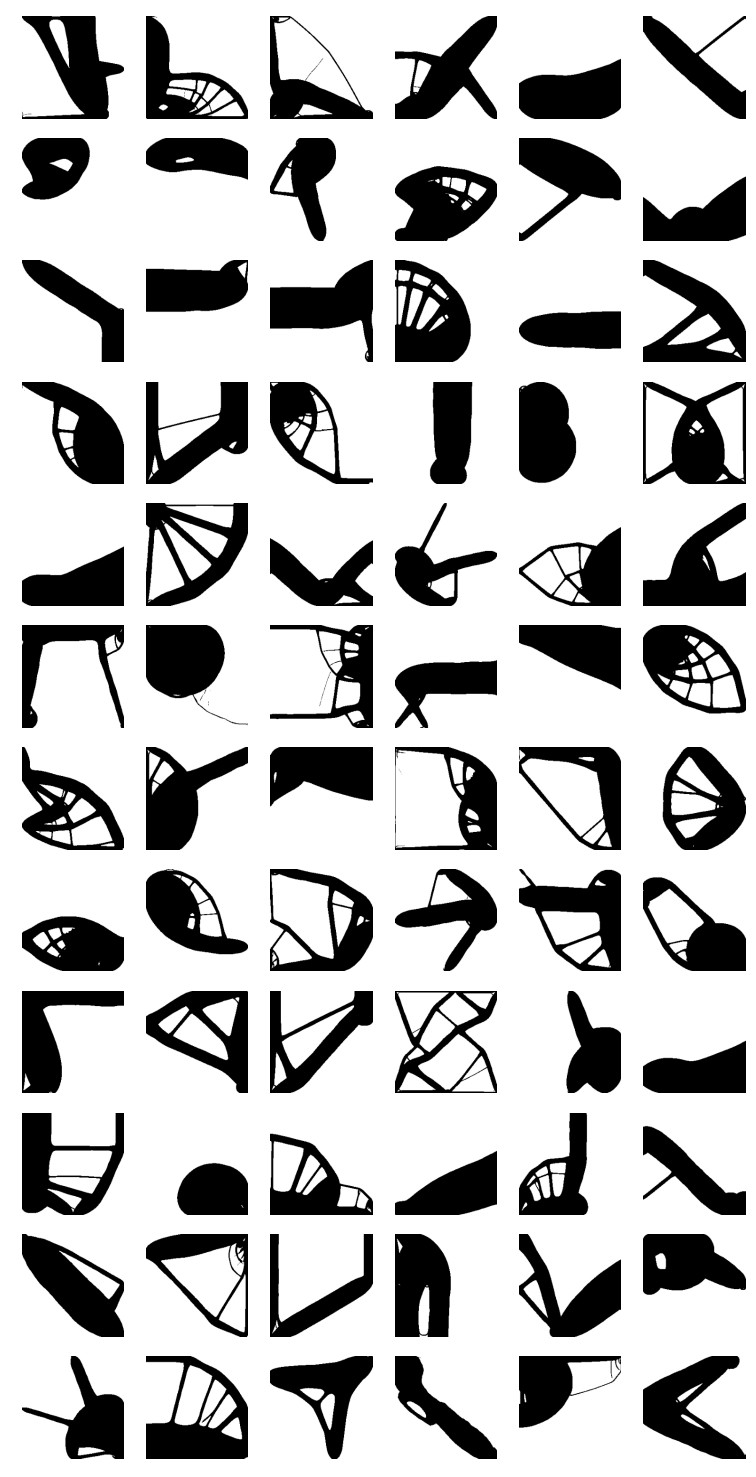

**Figure 18:** NITO generated topologies using a model trained on 64x64. Tested on the 256x256 data.

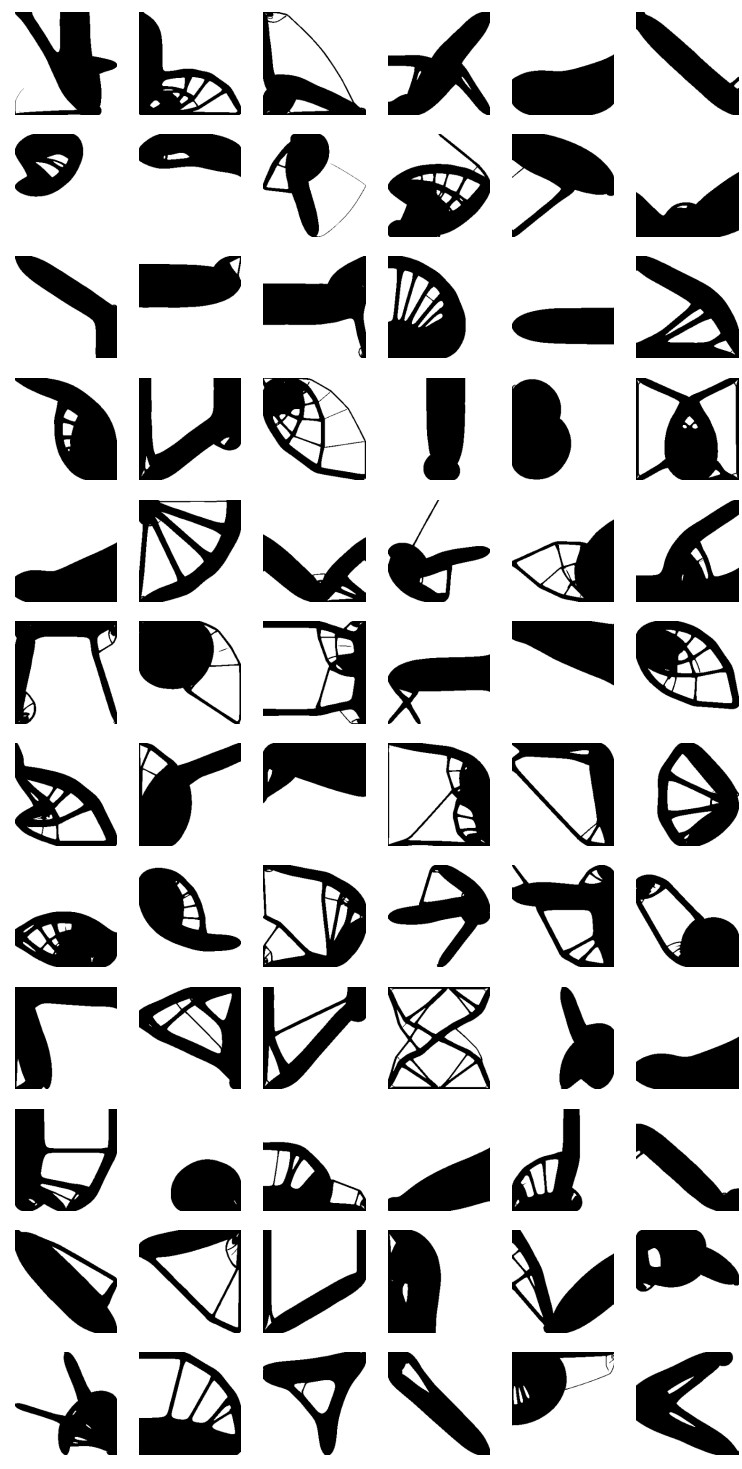

**Figure 19:** NITO generated topologies using a model trained on both 64x64 and 256x256. Tested on the 256x256 data.

# D  Dataset & Optimizer Details

Our experiments use a dataset of SIMP-optimized topologies in a unit square domain with a size of 64x64 and 256x256 and rectangular domains with different aspect ratios including 64x48, 64x32, and 64x16. For each topology in the dataset, information about the loads, supports, and volume fraction are included. Furthermore, we include the stress and strain energy fields. Our dataset is similar to the one proposed by Mazé & Ahmed (2023), however, we noted that the dataset used in prior works has been generated using an older version of SIMP, namely ToPy (Hunter et al., 2017).

This solver, although a robust implementation of SIMP, does not implement the latest improvements to the SIMP algorithm and uses a slower solver, which causes two issues. Firstly, the topologies that are used in the dataset proposed by Mazé & Ahmed (2023) are lower-performing topologies in comparison to what the latest solvers produce, hence possibly overestimating the performance of these models in some cases. Secondly, the SIMP method itself which prior studies compared their inference time with, did not use the fastest solvers, making those comparisons also somewhat inaccurate. To ensure that this is not the case in our studies,s we implement the SIMP optimizer from scratch in Python (The code for which will be publicly available), which performs the optimization using the latest and fastest implementation of the SIMP algorithm as far as the authors are aware (Wang et al., 2021b).

As such, we re-create the 64x64 dataset proposed by  Mazé & Ahmed (2023) using our solvers and find that the resulting topologies using our method are significantly better performing in comparison to the prior datasets used by many other works of research (Giannone et al., 2023; Giannone & Ahmed, 2023; Mazé & Ahmed, 2023; Nie et al., 2021b). Given this, it is safe to assume that the performance of the models in prior studies that compare to the inferior dataset may have been overestimated while also training models on lower-performance samples. However, to allow for a fair comparison, we retrain the best-performing model in the literature TopoDiff (Mazé & Ahmed, 2023), on our new dataset and rerun their experiments on this new dataset. However, we report the performance of other models as the original authors measured them overestimated or otherwise.

The 64x64 dataset includes 48,000 training samples and 1,000 test samples, which we use for testing our models. The 256x256 dataset includes 60,000 training samples and 1,800 samples for testing, and the other three domains have 29,000 samples each with 1,000 test samples each. The figures that follow visualize some of the training samples for each dataset. In the following figures, we include figures showing samples from all datasets.

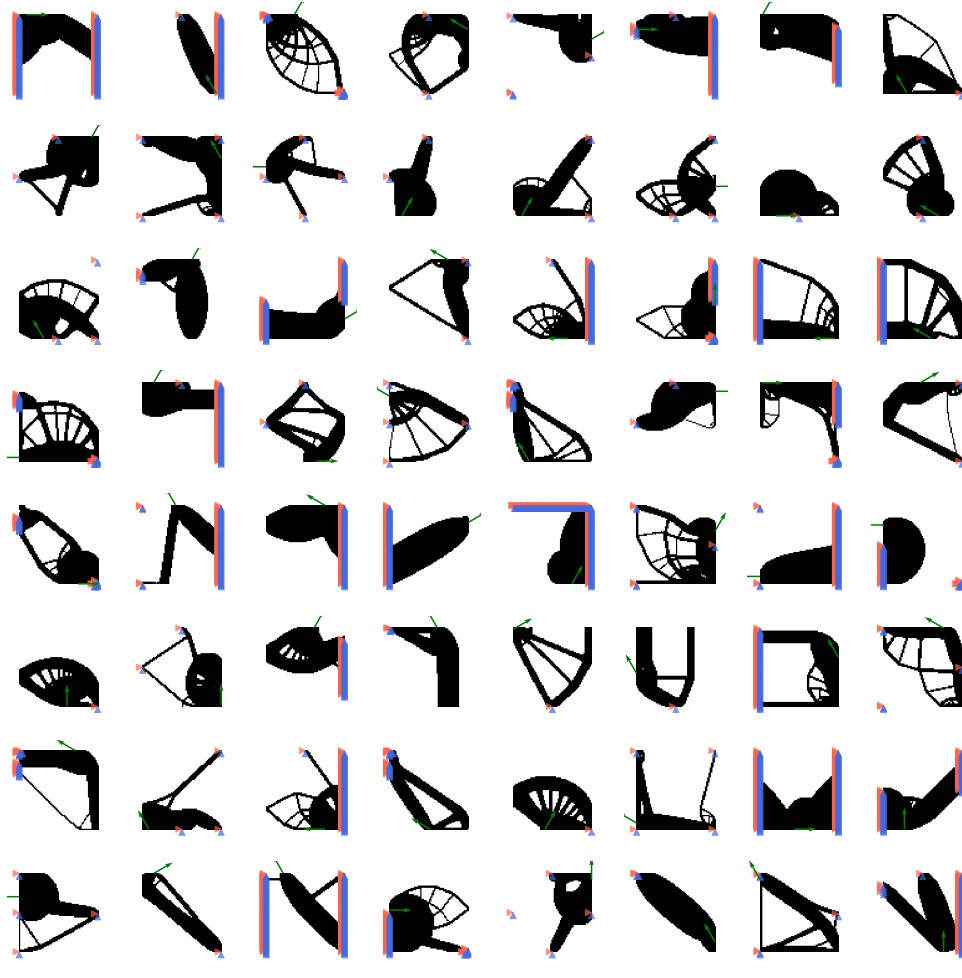

Constrained in Y    Constrained in X    Applied Load

**Figure 20:** Random samples from the 64x64 dataset. Green arrows show the locations and directions of the loads applied in each problem. Blue triangles indicate points that are constrained in x and red triangles indicate points that are constrained in y.

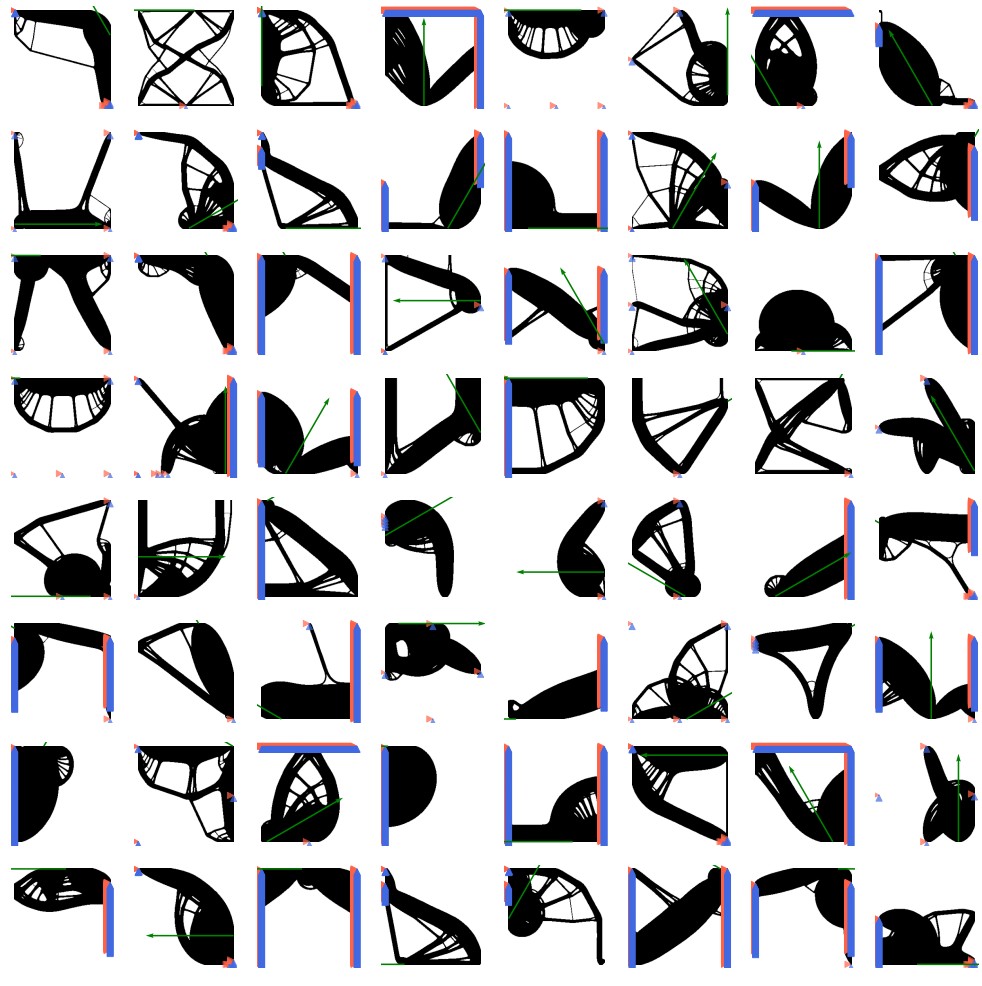

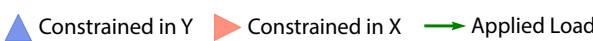 Constrained in Y ▶ Constrained in X → Applied Load

**Figure 21:** Random samples from the 256x256 dataset. Green arrows show the locations and directions of the loads applied in each problem. Blue triangles indicate points that are constrained in x and red triangles indicate points that are constrained in y.

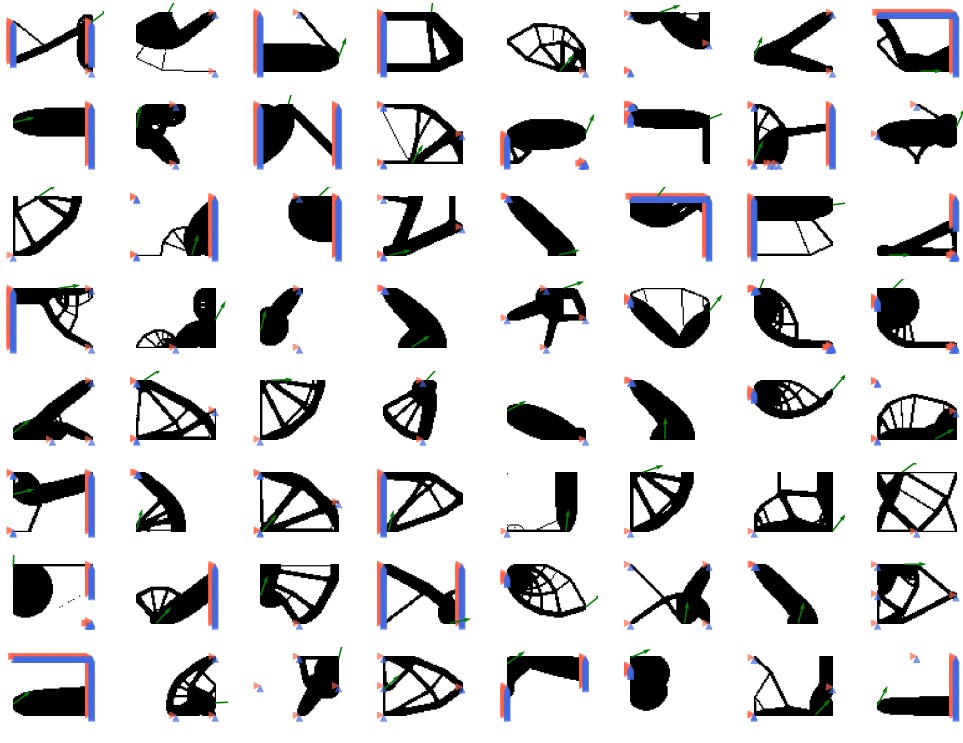

Constrained in Y ▶ Constrained in X → Applied Load

**Figure 22:** Random samples from the 64x48 dataset. Green arrows show the locations and directions of the loads applied in each problem. Blue triangles indicate points that are constrained in x and red triangles indicate points that are constrained in y.

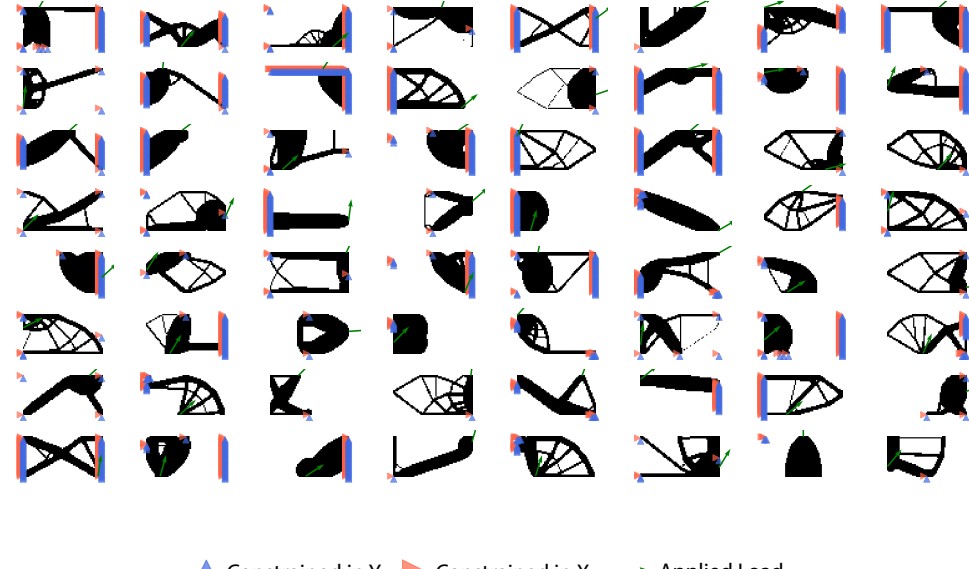

Constrained in Y ▶ Constrained in X ➝ Applied Load

**Figure 23:** Random samples from the 64x32 dataset. Green arrows show the locations and directions of the loads applied in each problem. Blue triangles indicate points that are constrained in x and red triangles indicate points that are constrained in y.

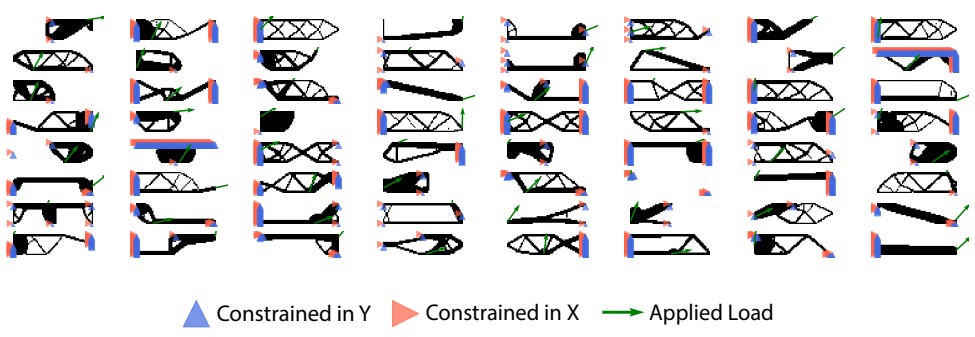

Constrained in Y ▶ Constrained in X ➝ Applied Load

**Figure 24:** Random samples from the 64x16 dataset. Green arrows show the locations and directions of the loads applied in each problem. Blue triangles indicate points that are constrained in x and red triangles indicate points that are constrained in y.

# E    Sampling At 25 Million Pixels

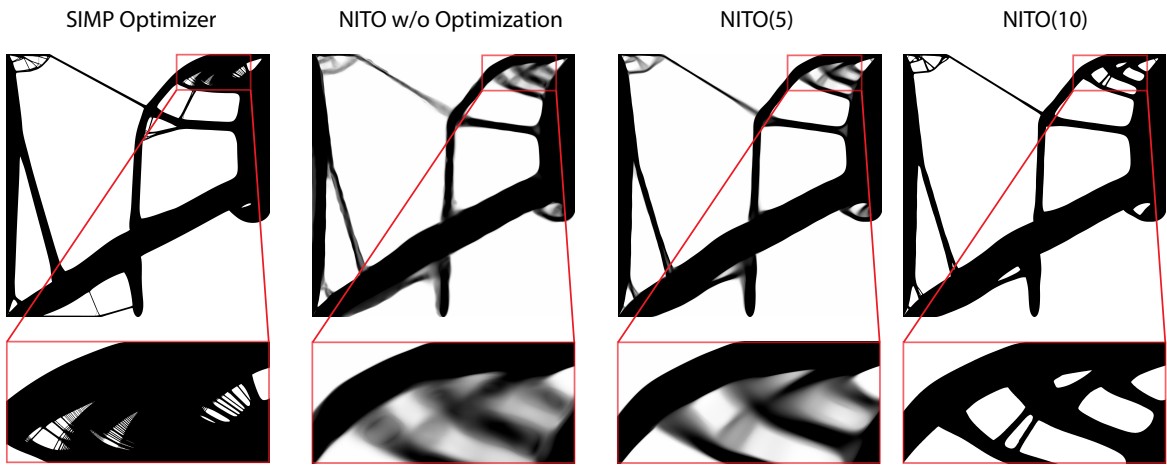

**Figure 25:** Comparing SIMP at native resolution of 5000x5000 (25 Million Elements) to NITO with and without direct optimization. We see that with optimization, the quality of samples increases, however, given NITO has not seen any data at this resolution, the solution still lacks very fine detail compared to the native resolution solution of the optimizer. Regardless as we saw in Table 6 these topologies perform very well despite the lack of very fine grained features.

As we mentioned, we tested NITO at much higher resolutions and observed a lower performance compared to resolutions in the dataset. This is because NITO is not trained at these resolutions and simply lacks the fine-grained details needed at these resolutions. However, we saw that NITO is capable of scaling efficiently to very high detail sampling and even with less detail a few steps of optimization brings performance close to the native resolution solution from the optimizer. However, here we present Figure 25 which shows that the issue of lacking detail persists even with a few steps of direct optimization. This means that as suspected data at very high resolution is needed to truly scale to such highly detailed topologies. In future works, we will be exploring how fine-tuning on few-shot examples at higher resolutions impacts this kind of experiment.

