# OpenReview forum: "NITO: Neural Implicit Fields for Resolution-free and Domain-Adaptable Topology Optimization"
_TMLR — Accepted by TMLR_

### Review · Reviewer_TmN7 · 2025-02-15

**Summary Of Contributions:**

The paper presents a neural network model for topology optimization (TO) problems with minimum compliance objectives. Specifically, the model maps the topology optimization problem setup (e.g., load and boundary conditions) to a neural-implicit representation of material distribution in the domain, which is further fine-tuned by standard SIMP optimization. The paper demonstrates its efficacy by comparing it with existing neural TO methods on 2D TO examples at up to 256 x 256 grid resolution.

The main contribution of this work includes the network model itself and the experiments that confirm its efficacy.

**Audience:**

Yes

**Broader Impact Concerns:**

I don’t see any concerns in this category.

**Claims And Evidence:**

Yes

**Requested Changes:**

- I suggest the paper adds reviews of and discussions about classic TO methods to position this work among them more properly. [1, 2] and their related works could be a good starting point. In particular, I suggest the paper reconsiders its claims about SIMP’s and this work’s pros and cons on large-scale TO problems.
- It is not obvious to me that Eq. 4 is “the same as the output of the SIMP optimizer.” It would be great to provide a derivation of it.
- “Notably, solving the linear system of the equations scales cubically (O(n3 )) with the number of nodes.” This seems a bit misleading. O(n^3) is the time complexity of solving a dense linear system naively, but K is typically a sparse and (semi-)positive symmetric matrix. In practice, TO solvers can benefit from conjugate gradient, multigrid, or [1] that are much faster than O(n^3).

**Strengths And Weaknesses:**

The paper is very well-written and easy to follow. I enjoyed reading it and did not spot major technical issues. Incorporating a resolution-free, neural-implicit surface representation into the standard topology optimization problem is a valuable proposal. Although some recent works have tackled similar settings before, the idea is still timely and can potentially spark some interesting downstream applications.

I feel the major weakness of this work is that the problem size (2D with 256 x 256 grid resolution) is relatively small compared with what classic TO methods tackle, e.g., giga-voxel 3D grids in [1, 2]. A neural implicit surface representation supports “infinite” grid resolution, and a sweet spot of such representation would be their potential to express delicate geometric details that would have needed large numbers of voxels in traditional grid-based representation. However, the proposed network seems to capture only relatively coarse geometric features. I can agree that this work seems superior to existing deep-learning-based TO methods. Still, their benefits over classic TO methods in terms of speed, optimality, and expressiveness on larger-scale problems seem unclear.

[1] Liu et al “Narrow-Band Topology Optimization on a Sparsely Populated Grid.” ACM Transactions on Graphics 2018.

[2] Aage et al “Giga-voxel computational morphogenesis for structural design.” Nature 2017.

---

> ### Author Response · Authors · 2025-03-11
> **Rebuttal Part 1**
>
> > The paper is very well-written and easy to follow. I enjoyed reading it and did not spot major technical issues. Incorporating a resolution-free, neural-implicit surface representation into the standard topology optimization problem is a valuable proposal. Although some recent works have tackled similar settings before, the idea is still timely and can potentially spark some interesting downstream applications.
> >
>
> We thank the reviewer for the positive feedback and the time and effort you devoted to review our work. We hope that our response addresses your concerns.
>
> > I feel the major weakness of this work is that the problem size (2D with 256 x 256 grid resolution) is relatively small compared with what classic TO methods tackle, e.g., giga-voxel 3D grids in [1, 2]. A neural implicit surface representation supports “infinite” grid resolution, and a sweet spot of such representation would be their potential to express delicate geometric details that would have needed large numbers of voxels in traditional grid-based representation. However, the proposed network seems to capture only relatively coarse geometric features. I can agree that this work seems superior to existing deep-learning-based TO methods. Still, their benefits over classic TO methods in terms of speed, optimality, and expressiveness on larger-scale problems seem unclear.
> >
>
> The reviewer's concern about resolution limitations is reasonable, given the presentation of results in the paper and the fact that classic TO is successfully applied to much larger problems. However, we would like to emphasize that NITO can be sampled at any resolution regardless of its training data. While this may mean that it lacks fine-grained features when sampled at much higher resolutions than its training data, even seeding classic TO with a very rough shape can significantly accelerate convergence. We agree that sampling at higher resolution here would not yield benefits since NITO was not trained on very high resolution data. However, this is not a limitation of the method, rather a limitation of the data. We have added a paragraph to our final section discussing this limitation and the potential for fine-tuning on a few very high resolution samples and exploring if a few-shot fine-tuning can be exploited in this context for further expansion of the work to very high resolutions.
>
> Most important to note is that NITO as a framework scales much better than conventional solvers, and to demonstrate this, we conduct a limited experiment on NITO at a much higher resolution of 5000x5000 (25 M elements). We see that NITO can be sampled at this kind of resolution very effectively and with much better scalability compared to conventional solvers. In fact, it performs very well despite the fact that we use a multi-grid ***GPU*** accelerated solver in these experiments.  Although the NITO-generated topology at high resolutions lacks the fine-grained details (as visualized in Appendix E), even with few steps of optimization, NITO is nonetheless able to achieve very small (<1%) compliance error with the few-step optimization. Although its performance is not perfect, we feel that this experiment showcases the framework's ability to scale efficiently to higher resolutions. Should a significant amount of higher resolution data be available, we have no doubt that performance would improve. Moreover, NITO's is completely resolution agnostic and scales regardless of the resolution of the underlying data, which makes it much better suited to such tasks and we highlight this in our conclusions. We hope that these experiments and the added context in our limitations address this concern.

---

> ### Author Response · Authors · 2025-03-11
> **Rebuttal Part 2**
>
> > It is not obvious to me that Eq. 4 is “the same as the output of the SIMP optimizer.” It would be great to provide a derivation of it.
> >
>
> We thank the reviewer for pointing out the flaw in the language. We agree that our intent with this language is easily misinterpreted. What we are trying to say is that $\rho(\mathbf{x} | \mathbf{C})$, the probability of material at $\mathbf{x}$ comes from the ground truth in the data which is derived from running SIMP in our dataset. We have updated this sentence to prevent this misunderstanding for the readers.
>
> > I suggest the paper adds reviews of and discussions about classic TO methods to position this work among them more properly. [1, 2] and their related works could be a good starting point. In particular, I suggest the paper reconsiders its claims about SIMP’s and this work’s pros and cons on large-scale TO problems.
> “Notably, solving the linear system of the equations scales cubically (O(n3 )) with the number of nodes.” This seems a bit misleading. O(n^3) is the time complexity of solving a dense linear system naively, but K is typically a sparse and (semi-)positive symmetric matrix. In practice, TO solvers can benefit from conjugate gradient, multigrid, or [1] that are much faster than O(n^3).
> >
>
> We are happy to expand our discussion of classic TO methods to provide more adequate context of our work, highlighting in particular the huge difference in resolutions between current AI-based TO and classic TO. We have done so in section 2.1.
>
> The main point being made in this paragraph is not to bring down conventional optimizers or solvers. This is meant to point to the reality that solving a linear system of equations requires more than linear computational complexity in comparison to our proposed method. When running NITO one samples once for each element $O(n)$ while when solving a linear system, say in 2D, one must solve a system with size $\approx 2n$-by-$2n$, which will be rather slow even with multi-grid solvers. This is because in a multi-grid setup that is "ideal" the best case complexity is thought to be $O(N=2n\approx n)$ for a sparse system of equations, however every matrix vector product that is computed has cost $O(N=2n \approx n)$, and in most multi-grid setups the solution is iterative anyways and not easily parallelized. This is not true with neural fields, not only are they sampled in parallel, the computational cost scales linearly with increased designs, while multi-grid convergence will differ based on the spectral radius of the underlying matrix and system. However, we believe the point brought up is fair, especially since NITO also utilizes a few steps of direct optimization. As such, we have adjusted the language here to point out the reality that in larger problems more optimized methods exist for solving the system of linear equations. However, we also still emphasize that the computational cost does not always scale linearly in conventional methods and their iterative nature makes them slower in general. We hope that this has addressed the reviewer's concerns in this regard.

---

### Review · Reviewer_idtH · 2025-02-17

**Summary Of Contributions:**

In this paper, the authors propose a learning-based topology optimization method called NITO. At its core, the method introduces three ideas to improve topology optimization in terms of speed and quality compared to existing approaches: 1) the proposed Boundary Point Order-Invariant MLP (BPOM) approach effectively integrates conditioning for the predicted shapes, such as boundary conditions and loads, 2) the authors use a neural implicit field representation, enabling NITO to synthesize topologies of any shape or resolution, and 3) finally, we introduce an inference-time refinement step using a few steps of gradient-based optimization, allowing NITO to achieve results comparable to direct optimization methods.

**Audience:**

Yes

**Claims And Evidence:**

Yes

**Requested Changes:**

* [Strengthen] Section 1: Replace “deep learning issues have struggled with several issues” with “deep learning approaches have struggled with several issues.”

* [Strengthen] Section 2.1: Correct “folume” to “volume.”

* [Strengthen] Section 2.2: Extend the discussion of existing learning-based topology optimization approaches, including relevant works such as [1, 2, 3, 4, 5].

* [Strengthen] Consider discussing the neural operator [6] approach in the context of the considered task, specifically addressing why it was not used instead of neural fields.

* [Strengthen] Clarify how exactly the proposed method ensures the fulfillment of the imposed conditions. It would be beneficial to provide more details in the method and experimental sections.

**Strengths And Weaknesses:**

# Strengths

* The paper contributes to the field of learning-based physics methods, particularly topology optimization, which is steadily gaining attention.
* The authors provide a comprehensive introduction to topology optimization and modern existing approaches.
* The proposed method, while conceptually simple to implement, demonstrates significant improvements over other baselines.

# Weaknesses

* The paper primarily focuses on a specific type of topology optimization that aims to predict shapes based on a given set of conditions. Meanwhile, several works, such as [1, 2, 3, 4, 5], also operate within the domain of optimal shape generation under given conditions while simultaneously optimizing a physical property (e.g., aerodynamic drag).
* One of the method’s key contributions is the use of a neural implicit field representation for shape prediction, allowing for resolution-independent predictions. While this is a notable advantage, existing methods like Neural Operators [6] were designed precisely for this reason and are provably capable of generating predictions irrespective of resolution.
* Additionally, the proposed conditioning procedure does not ensure that the imposed conditions are satisfied in the final predictions. This issue could potentially be addressed with an additional SIMP iteration, but without it, the constraints remain unmet.

[1] Baque, Pierre, et al. "Geodesic convolutional shape optimization." ICML 2018.

[2] Durasov, Nikita, et al. "Debosh: Deep bayesian shape optimization." arxiv 2021.

[3] Durasov, Nikita, et al. "Enabling Uncertainty Estimation in Iterative Neural Networks." ICML 2024

[4] Viquerat, Jonathan, et al. "Direct shape optimization through deep reinforcement learning." Journal of Computational Physics

[5] Wei, Zhen, et al. "Diffairfoil: An efficient novel airfoil sampler based on latent space diffusion model for aerodynamic shape optimization." AIAA Aviation Forum 2024.

[6] Kovachki, Nikola, et al. "Neural operator: Learning maps between function spaces with applications to pdes." JMLR 2023

---

> ### Author Response · Authors · 2025-03-11
> **Rebuttal**
>
> Thank you for your thoughtful and detailed review of our paper. We sincerely appreciate your constructive feedback, which highlights both the strengths of our work and areas for improvement. Your suggestions, particularly regarding the discussion of existing approaches, the role of neural operators, and the fulfillment of imposed conditions, are extremely valuable and will help us strengthen our manuscript.
>
> We will carefully address each of your comments in our response below:
>
> > The paper primarily focuses on a specific type of topology optimization that aims to predict shapes based on a given set of conditions. Meanwhile, several works, such as [1, 2, 3, 4, 5], also operate within the domain of optimal shape generation under given conditions while simultaneously optimizing a physical property (e.g., aerodynamic drag). ...
> [Strengthen] Section 2.2: Extend the discussion of existing learning-based topology optimization approaches, including relevant works such as [1, 2, 3, 4, 5].
> One of the method’s key contributions is the use of a neural implicit field representation for shape prediction, allowing for resolution-independent predictions. While this is a notable advantage, existing methods like Neural Operators [6] were designed precisely for this reason and are provably capable of generating predictions irrespective of resolution. ...
> [Strengthen] Consider discussing the neural operator [6] approach in the context of the considered task, specifically addressing why it was not used instead of neural fields.
> >
>
> We agree that some of the works cited are analogous to the TO problem at hand in that they design a shape to optimize a physical property subject to constraints and physics considerations. While the setting in our work on direct optimization of continuum is different, we acknowledge that adding extra context from ML and ML-adjacent communities may be particularly valuable for a TMLR audience. As such, we are happy to include the suggested literature in accordance with the reviewer's suggestion. We have extended our background to also include neural operators and shape optimization using deep learning and have clearly highlighted how implicit neural fields are highly similar to neural operators in the context of continuum structural optimization. Note that here we are not optimizing shapes, rather distribution of material in a continuum. This means that even when function spaces are used, they are accompanied by a level-set operation and effectively reduce to field function spaces, where the structure is determined by threshholding the field. This is mathematically identical to an implicit neural field (which NITO uses). We hope that our added discussion on shape optimization and neural operators have addressed the reviewers concern.
>
> > Additionally, the proposed conditioning procedure does not ensure that the imposed conditions are satisfied in the final predictions. This issue could potentially be addressed with an additional SIMP iteration, but without it, the constraints remain unmet. ...
> [Strengthen] Clarify how exactly the proposed method ensures the fulfillment of the imposed conditions. It would be beneficial to provide more details in the method and experimental sections.
> >
>
> We acknowledge that as a primarily data-driven method, NITO has no explicit enforcement of the volume fraction constraint. However, by learning based on examples which satisfy these constraints, NITO does in fact learn to satisfy these constraints in the overwhelming majority of cases (albeit implicitly, rather than explicitly). As the reviewer notes, and as we demonstrate in the paper, minor constraint violation is easily resolved in a handful of SIMP iterations. With this said, adding more explicit constraint satisfaction (such as an explicit volume fraction penalty or a penalty for leaving voids at applied loads) should be simple to apply and could conceivably improve performance. Most importantly, we threshold the field at a value of $0.5$, which comes with no guarantee of meeting volume fraction requirements. However, a simple binary search on the field would allow for finding a threshold which ensures the volume fraction constraint is met. However, this is not part of the proposed pipeline. We have added a paragraph in the experimental setup section going over this distinction. If the reviewer considers such an experiment on evaluating NITO on a binary search determined threshold insightful, we are happy to add it in a later draft.
>
> > Section 1: Replace “deep learning issues have struggled with several issues” with “deep learning approaches have struggled with several issues.” ... Section 2.1: Correct “folume” to “volume.”
> >
>
> We appreciate these catches and have addressed these mistakes along with some other grammar and spelling issues.

---

### Review · Reviewer_uPWv · 2025-02-20

**Summary Of Contributions:**

The paper presents an implicit neural representation (INR) network for learning 2D topologies. The method combines SIREN with fourier expansions, and uses FiLM to introduce conditioning by the topology parameters. The topology constraints are handled by a PointMLP type network. The overall network combines multiple architectures that are all well-motivated and appropriate for the problem. Finally, the method is also shown to work well with simulator post-refinement, and achieve better performance than diffusion based method.

**Audience:**

Yes

**Broader Impact Concerns:**

No issues.

**Claims And Evidence:**

Yes

**Requested Changes:**

I am not familiar with this problem domain. The paper uses special terminology, which was not translated into common english. The math presentation is cursory and skips over details and definitions. The paper is likely easy to follow for a domain expert, but for an outsider the topic and material should be made more explicit and concrete.

Can you explain what are we looking at in figs1+2: at first this looks like ink blobs. Are these some physical objects, like buildings? These seem 2D: why is modelling things in 2D relevant?

The figures need to be properly annotated. For instance, there are triangles and arrows that are not explained. The figure contents should be labelled, and coupled with the math symbols as appropraite.

I’m a bit confused of the circularity of eq 1’s min(Fd) and Kd=F. This then becomes min Kd=d, which looks like an eigenvalue problem, but without the eigenvalues (ie. Kd=lambda*d). Can you give a bit more exposition on how should we interpret the Kd=d? Can you explain what K,F,D mean?

All math presentation and notation needs to be explicit, well-defined, self-evident and properly introduced. As an example, eq 1 is already confusing:
- The domains and sizes of the variables are not defined
- It’s not clear what the “design variables” mean, or what “problem domain” means. Can you give some helpful examples?
- The Fd reads like a matrix-vector product that results in a vector, but the symbol f is not boldface. Is the result then vector or scalar? If the result is vector, what does “min” mean over a vector?
- v is undefined
- I don’t understand the ^e notation: is this an exponent or some indexing notation?
- I don’t understand the basic terminology here, eg. “load”, “displacement”, “nodal”, “element”, etc.
- What is the coordinate space here?

The treatment of rho’s is quite confusing to me. We begin with rho(phi), then later have rho(x), then hat{rho}(x|C,theta), then rho(x|C). Why do you need so many rho’s, and what does each of them represent? What does the \hat, phi and theta mean as arguments of rho? In eq 3 we have theta on both sides: what does this mean?

The BPOM description is vague, and it should be similarly mathematical as in eqs 5+6. PointMLP already uses pooling, so how is your pooling strategy different and why?

**Strengths And Weaknesses:**

S: The method is principled, well-motivated and appropriate for the problem at hand.

S: The results are comprehensive and well-presented, and show the performance to be competetive without post-tuning, and SOTA with.

W: The presentation assumes special domain knowledge, and the setting is not opened up sufficiently for general ML audience.

---

> ### Author Response · Authors · 2025-03-11
> **Rebuttal Part 1**
>
> Thank you very much for your thoughtful and detailed feedback. Your comments will greatly help us improve the clarity and accessibility of our paper. Below, we address each of your concerns individually.
>
> > Can you explain what are we looking at in figs1+2: at first this looks like ink blobs. Are these some physical objects, like buildings? These seem 2D: why is modelling things in 2D relevant?
> >
>
> These are 2D physical structures designed to withstand specific forces. Such structures have numerous real-world use cases, ranging from bridges to jet engine mounting brackets. Many planar structural elements exist in practice (for example, sheet metal components). Additionally, many real-world 3D topology optimization cases exhibit symmetry in one dimension and can thus be effectively modeled in 2D. The resulting manufactured component would then become a linearly symmetric 3D part, obtained by giving depth to the 2D solution of the topology optimization problem. With that said, nothing inherently limits NITO to 2D problems, provided an appropriate dataset. Nevertheless, we acknowledge and support the reviewer's point that real-world problems are primarily 3D. NITO aims to develop the foundational methods upon which general topology optimization problems can be addressed. As mentioned, everything presented here can easily be extended to 3D if suitable data becomes available.
>
> > The figures need to be properly annotated. For instance, there are triangles and arrows that are not explained. The figure contents should be labelled, and coupled with the math symbols as appropraite.
> >
>
> Thanks for this feedback. The figures in the main body do include descriptions of symbols on the figure themselves (the second time they appear they are not annotated to make the figure easier to parse), but we recognize that the figures in the appendix do not have legends and only mention symbols in the caption. We have added legends to all such figures. We hope that this has satisfied the reviewer's concerns.
>
> > I’m a bit confused of the circularity of eq 1’s min(Fd) and Kd=F. This then becomes min Kd=d, which looks like an eigenvalue problem, but without the eigenvalues (ie. Kd=lambda*d). Can you give a bit more exposition on how should we interpret the Kd=d? Can you explain what K,F,D mean?
> >
>
> We thank the reviewer for pointing out the potential for misunderstanding in the optimization formulation. The paragraph following Equation 1 does include details on what each term means. Equation 1 is not circular, note that the equality constraint $K(\phi) d=F$ and the objective $F^T d$ are not interchangeable as suggested. Minimizing the dot product $F^T d$ is not equivalent to the equation $K d=d$, which would indeed represent an eigenvalue problem. Instead, the objective and the equality constraint in this formulation can be combined into an objective of the form $F^T K(\phi)^{-1} F$, subject only to linear inequality constraints. Note here that $K^{-1}$ is distinct from $K$, which clearly differentiates the problem from an eigenvalue problem.
>
> To clarify this further, we have revised the paragraph preceding this equation, explicitly highlighting the following:
>
> - $F$ is the right-hand side vector representing the loads applied to the problem domain. This vector is specific to each scenario, including the locations and directions of the applied loads. We have now used a lowercase $f$ consistently to better reflect that it is a vector.
> - $K(\phi)$ is a symmetric positive definite (SPD) matrix, thus invertible and full rank-that depends nonlinearly on the design variables $\phi$ and the boundary conditions of the problem.
> - $d$ is the displacement field defined at the nodes of the discretized domain, i.e., the mesh used in the numerical solution.
>
> We hope these clarifications address the reviewer's concerns and have edited the paragraph preceding Equation 1 accordingly.

---

> > ### Comment · Reviewer_uPWv · 2025-03-12
> >
> > Thanks for the response. This clarified a lot. I still see vague presentation, which needs to be explicit and precise.
> >
> > - Sec 2 talks about supports, loads and forces, but eq 1 only contains loads out of these. What are then forces and supports?
> > - In figs the two types of blue triangles are unclear. The singleton force arrow along boundary is baffling. Why along the boundary? Why only one force? I'm struggling to understand this: it looks like torque..
> > - It's not clear what are the sizes of the variables
> > - It's not clear how the variables depend on each other. For instance, $d$ is written as free variable without dependencies, but it seems to be a function of f and K.
> > - It's not clear what variables are fixed inputs
> > - The notation in appendix is different from main paper (u vs f)
> > - In appendix u is a vector field, and in main paper f is just a "vector". Later in paper the forces seem to be a point cloud (eg. a matrix).
> > - d is not defined in main text: I think it's a vector field
> > - Can you clarify what are inside $C$ in eq 3? It seems that C does not know about forces or supports: why not? I again wonder what are the inputs to the whole system.
> > - Fig 3 has "constraint points", which are not part of eq 1 or fig 2. What are these?

---

> ### Author Response · Authors · 2025-03-11
> **Rebuttal Part 2**
>
> > I am not familiar with this problem domain. The paper uses special terminology, which was not translated into common english. The math presentation is cursory and skips over details and definitions. The paper is likely easy to follow for a domain expert, but for an outsider the topic and material should be made more explicit and concrete.
> >
>
> > All math presentation and notation needs to be explicit, well-defined, self-evident and properly introduced. As an example, eq 1 is already confusing:
> The domains and sizes of the variables are not defined
> It’s not clear what the “design variables” mean, or what “problem domain” means. Can you give some helpful examples?
> The Fd reads like a matrix-vector product that results in a vector, but the symbol f is not boldface. Is the result then vector or scalar? If the result is vector, what does “min” mean over a vector?
> v is undefined
> I don’t understand the ^e notation: is this an exponent or some indexing notation?
> I don’t understand the basic terminology here, eg. “load”, “displacement”, “nodal”, “element”, etc.
> What is the coordinate space here?
> >
>
> We thank the reviewer for highlighting the potential confusion in terminology and mathematical presentation. To address these concerns, we have added a comprehensive appendix containing a detailed derivation of the minimum compliance topology optimization (TO) problem, along with a clear, tangible description of the terminology used throughout the paper. We believe this appendix will make the concepts more accessible to readers outside the immediate problem domain.
>
> We also recognize the importance of clearly defining terminology and mathematical symbols. We have adjusted the Figure 2, emphasizing that the domain in a TO problem is the region of space over which the problem is solved. Specifically, we now clearly indicate that the square shown in Figure 2 represents a physical 2D domain, and the design variables are represented as a density field within this domain. The symbol $\Omega$ has been explicitly added to the figure, and we have introduced clearer explanations in both the caption and the surrounding text regarding design variables. We've clarified that the density field shown visually in Figure 2 corresponds directly to the "design variables" discussed in the mathematical formulation.
>
> Regarding the other points raised:
>
> - $V$ : The sentence preceding the equation explicitly defines $V$ as the volume fraction constraint. We have verified that this definition is clear and explicitly introduced in the revised text.
> - $^e$ : The notation $^e$ refers to an element $e$ in the discretized domain (mesh). We have explicitly clarified this in the revised paragraph. The exact number of design variables is not fixed and depends on the resolution of the mesh. Indeed, handling varying resolutions without a fixed number of design variables is one of the main challenges addressed by NITO. We have added more context in the paragraph before equation 1, which we hope makes this clear from the start.
>
> **Description of Terminology (For The Reviewer):**
>
> - Load: A force applied to some region of the problem domain.
> - Displacement: The deformation or shift experienced by the structure from its unloaded equilibrium when a load is applied. The displacement quantifies the extent and direction of structural deformation under load.
> - Nodes and Elements: Numerical solutions to PDEs using finite elements involve discretizing the domain into a mesh. The resulting discrete points are "nodes," and the subdivisions of the mesh connecting these nodes are called "elements." Both terms have been defined accordingly in the revised text.
>
> We have stated these clarifications in the newly added appendix, aiming to fully address the reviewer's concerns. The added explanations and detailed derivations in the appendix should help alleviate ambiguity and make the material more concrete and accessible to readers outside the immediate problem domain.

---

> ### Author Response · Authors · 2025-03-11
> **Rebuttal Part 3**
>
> > The treatment of rho’s is quite confusing to me. We begin with rho(phi), then later have rho(x), then hat{rho}(x|C,theta), then rho(x|C). Why do you need so many rho’s, and what does each of them represent? What does the \hat, phi and theta mean as arguments of rho? In eq 3 we have theta on both sides: what does this mean?
>
> All mentions of $\rho$ in different contexts are consistent. Note that when formulating the optimization problem $\rho$ is a function of the design variables. The added appendix should make this clear as well. Later the context of the problem shifts from the conventional optimization to the context of predicting the densities (not the design variables) directly. Since NITO is a neural field we must formulate $\rho$ as a ***field*** in space hence making $\rho$ a function of coordinates. Finally, we have contextualized densities in the context of the machine learning setup. Note that the neural field background makes it clear that a neural field is a function of coordinates $\mathbf{x}$ and the neural network parameters $\Theta$. Finally, for different load boundary conditions and volume fraction, the distribution of densities changes hence making it conditional on the latent constraint representation $\mathbf{C}$. Thus, we believe that equation 3 is fairly clear. Nonetheless, we have changed the language in the paragraph that follows the equation to make this more immediately evident. The $\hat{}$ on $\rho$ is added to clearly distinguish this ***prediction*** by the neural network from the ground truth from optimization denoted by $\rho$. We believe that these notations are needed to arrive at the objective of the neural network and are consistent in the context of the paper. If the reviewer is not satisfied by this explanation, we would appreciate it if the reviewer can clarify which part of the notation is confusing, and we will adjust the language in future drafts.
>
> > The BPOM description is vague, and it should be similarly mathematical as in eqs 5+6. PointMLP already uses pooling, so how is your pooling strategy different and why?
>
> To address this, we have further edited the description of BPOM and provided additional elaboration in the updated figure. Since pooling and concatenations are not easily parsed in an equation form when reading, we believe the better way to clarify this is through additional visual details on how the layer works. If the reviewer feels strongly, we would be happy to add a mathematical description in a later draft. Regarding PointMLP, we do not utilize geometric affine grouping and transforms but simply use an MLP and perform min, max, and average pooling operations and concatenate the results. We found that the more expensive and complex parts of PointMLP are not needed in this case. As such we use a simpler architecture. We hope that the added figure clarifies this and addresses the reviewer's concerns.

---

> ### Author Response · Authors · 2025-03-15
> **Response To Reviewer Part 1**
>
> We thank the reviewer for the additional feedback. It seems we could benefit from a more simple description of the TO problem in layman's terms. We have updated our exposition to the following:
>
> "In structural applications, TO often aims to minimize compliance (structural deformation) given a set of force loads,  supports, and a material volume limit. These loads, supports, and volume limit (`volume fraction') are casually referred to as the problem constraints, which are illustrated in Fig. 2. The goal in TO is to find an optimal structural layout (topology), given the problem constraints. Optimality is calculated as the compliance (deformation per unit force applied) of the structure when subjected to the applied loads and when the supported locations of the structure are constrained not to move."
>
> As seen above, we have also streamlined some terminology to avoid confusion. Miscellaneous terminology, such as "boundary conditions," "constraint points," "forces", "loads" "constraints", "supports", "volume ratio", etc. have been unified under common terminology and are now called "loads", supports" and "volume fraction". The collection of these is referred to casually as the problem constraints.
>
> In general, we have been intentional about omitting extensive details from the main paper that we feel are irrelevant to a general reader's comprehension of our contributions. We subsequently move such details to the Appendix. If the reviewer feels strongly that these must be included in the main paper, we are open to making appropriate adjustments in a future revision. However, we also feel that several of the reviewer's critiques are directly addressed in the existing draft.
>
> > Sec 2 talks about supports, loads and forces, but eq 1 only contains loads out of these. What are then forces and supports?
> >
>
> On forces we have adjusted the language to unify this with loads. Forces and loads are interchangeable in this context. On the equation not involving supports/constraints, we believe we have clarified these in the appendix. Specifically, in section A.2 we have this explanation:
>
> "Note that solving the linear elasticity PDE without any boundary conditions is not possible, as such, when performing FEA we will also set the boundary condition of a given problem. In the case of minimum compliance TO the boundary conditions are simply the locations of the supports~(i.e., points in space where the structure is fixed and not allowed to move. These supports can fix a set of nodes in the mesh and dictate no movement in all or any directions (see Figure 2). Such supports are introduced to the system by adjusting the stiffness matrix in FEA."
>
> That is to say, when a support prevents displacement, the stiffness matrix K on that row and corresponding columns will be adjusted in FEA to force this. we believe this is self-evident and should be clear that supports do not explicitly appear in any of the equations of optimization.
>
> > In figs the two types of blue triangles are unclear. The singleton force arrow along boundary is baffling. Why along the boundary? Why only one force? I'm struggling to understand this: it looks like torque..
> >
>
> We hope that our updated exposition helps address the confusion here. To answer this specific set of questions: There is one type of triangle that indicates a constraint on one degree of freedom and one triangle that indicates a constraint on two. These are commonly accepted in mechanics to indicate supports/constraints. Since we don't expect a mechanics background, the Figure is explicitly labeled (e.g. "Fixed in X"). This is just a mere example problem where the force happens to be along the edge of the problem and is applied purely in the "X" direction. It is not a torque. TO problems can have multiple applied forces but our dataset only features problems with one force and we apply these forces on the boundaries as these in most real-world application of TO of structural simulations body forces and internal forces are not common (Note that benchamrk problems in TO namely the cantilever bean and MBB beam and the bridge problem all only have loads at the boundaries. Discussing these nuances would require the paper to go over a handbook of numerical methods in structures and FEA). In our dataset (which is an extension of the data used in prior SOTA), which we visualize examples from in the appendix, you can see that the force can be applied in different locations and with different magnitudes. Therefore the supports are already labeled and the single force on the boundary is nothing baffling or unconventional (note that in 2D we are assuming the problem represents a symmetric shape or a thin plate, in the former the singleton load will be distributed along an extrusion and in the latter a singleton load is not meaningless since, for example, a rope attached to a plate or a contact force applied by to the plate by a finger etc is approximated as a single force).

---

> ### Author Response · Authors · 2025-03-15
> **Repone to Reviewer Part 2**
>
> > It's not clear what are the sizes of the variables
>
> The number of variables is explicitly discussed in the new draft. The paragraph before equation 1 explicitly mentions:
>
>
> "Note that the number of design variables is dictated by the number of elements in $\Omega$, which depends on the underlying discretization, and mesh. In this paper, we use structured meshes in rectangular domains, and refer to resolution instead of element counts. a more in-depth discussion on the topic and comprehensive explanation of the optimization problem are provided in Appendix A.
> "
>
> It should be clear that the design variables are the densities at each element(pixel) and one of our main contributions here is that the same model can be applied to different resolutions, hence different numbers of design variables. We believe the change we made in addressing this concern in the prior response has already addressed this. If the reviewer has a specific change in mind that will address this better, we are happy to make further clarifications on this matter in a way that the reviewer believes would be more illuminating to the audience of the paper.
>
> > It's not clear how the variables depend on each other. For instance, $d$ is written as free variable without dependencies, but it seems to be a function of f and K.
> >
>
> Free auxiliary variables are very common in any optimization problem, be it linear programming, mixed-integer linear programming, quadratic and conic programming or non-linear problems like ours. This is the same procedure as virtually any other optimization formulation. $d$ is free to change except that the equality constraint $K(\phi)d=f$ will dictated exactly it’s values given design variables. As equality implies this will mean that d will be the solution of this linear system of equations, which as we describe in appendix A, is non-linearly dependent on design variables, $\phi$, which again, as discussed in appendix A, ultimately define the density of material in the domain of the problem. When optimizing, one would obtain the gradients of the objective with respect to design variables, update said variables, and determine, implied by the equality constraint, the updated values for $d$. Thus $d$ depends on $f$ and $K$ (which is dependent on the design variables and supports) during the Finite Element Analysis. This is extensively discussed in Appendix A, which makes the distinction between the FEA and optimization step explicit.
>
> > It's not clear what variables are fixed inputs
> >
>
> We are quite explicit about the problem definition: "Fig. 2 illustrates the common structural TO problem formulation, in which a set of forces and supports are specified, along with a target volume fraction."
>
> This is unchanged from the previous draft.
>
> > The notation in appendix is different from main paper (u vs f)
> In appendix u is a vector field, and in main paper f is just a "vector". Later in paper the forces seem to be a point cloud (eg. a matrix).
> >
>
> We are very explicit in connecting the notations of the appendix to the paper. $f$ does indeed appear exactly in the appendix and has nothing to do with u. $u$ is essentially the same as $d$ except the fact that $d$ is the flattened version of $u$. We explicitly say this in the appendix:
>
> "where $\mathbf{d}$ is the displacement vector~(can be assembled into the displacement vector field $\mathbf{u}$ in the original PDE) which determines the displacement in each node of a given mesh and $\mathbf{f}$ is the force applied to the structure."
>
> We don't think there is any notation inconsistency here whatsoever.
>
> > d is not defined in main text: I think it's a vector field
> >
>
> From the text: "The objective is to minimize the compliance $\mathbf{f}^T \mathbf{d}$. $\mathbf{f}$ is the load vector and $\mathbf{d}$ the nodal displacement derived from $\mathbf{K}(\phi) \mathbf{d}=\mathbf{f}$, where $\mathbf{K}(\phi)$ is the stiffness matrix contingent on the design variables $\phi$."
>
> $d$ is the flattened vector field see Appendix A and above.
>
> > Can you clarify what are inside in eq 3? It seems that C does not know about forces or supports: why not? I again wonder what are the inputs to the whole system.
> >
>
> Based on the reviewer's confusion here, we have concluded that the meaning of "boundary conditions" was not made clear in the manuscript. Thus, we have removed this terminology in favor of explicitly mentioning "loads and supports" in the relevant areas. The explanation for Eq. 3 now reads:
>
> $\mathbf{C}$ is a condition vector that includes information about the domain shape, loads, supports, and desired volume ratio.
>
> We hope this is now amply clear.
>
> > Fig 3 has "constraint points", which are not part of eq 1 or fig 2. What are these?
> >
>
> As we mentioned above, we have unified the terms and Figure 3 now reads X supports and Y supports.

---

### Author Response · Authors · 2025-03-11
**Response to reviewers**

We sincerely appreciate the time and effort that the reviewers have put into evaluating our work. Their constructive feedback has been invaluable in helping us improve the clarity and presentation our paper. Below, we summarize the key revisions we have made based on the provided suggestions:

1. **Clarification of Figures and Notation:** We have improved figure annotations, added legends where necessary, and clarified mathematical symbols and notation, particularly around Equation 1 and the definition of design variables.

2. **Expanded Mathematical Exposition:** We have revised the explanation of our optimization formulation to eliminate potential misunderstandings, ensuring clear definitions of all terms, constraints, and objectives.

3. **Additional Background:** We have extended Section 2.2 to include a broader discussion of existing learning-based TO methods and neural operators, positioning our work more clearly in the broader context. We have also expanded section 2.1 to include a brief discussion on high resolution conventional solvers and how NITO is positioned in this regard.

4. **Clarified Conditioning and Constraint Satisfaction:** We have explicitly discussed how our method implicitly satisfies volume fraction constraints and suggested a simple post-processing step (binary search) to ensure strict adherence.

5. **Discussion on Scalability and High-Resolution Performance:** We have addressed concerns regarding problem size by running additional experiments at higher resolutions (e.g., 5000×5000). These results demonstrate NITO's ability to scale effectively, though we acknowledge limitations in capturing fine-grained details beyond its training resolution.

6. **Stronger Positioning Against Classic TO Methods:** We have refined our discussion on how neural implicit representations compare to classic TO solvers, particularly regarding computational complexity and scaling behavior.

7. **Clarifications on BPOM and Network Design Choices:** We have improved our explanation of BPOM’s pooling strategy and its distinction from PointMLP, supported by an updated figure for clarity.

8. **Grammar and Typographical Corrections:** We have fixed minor language issues throughout the manuscript to improve readability.

9. **Diff:** To make it easier to review the changes we have added a LatexDiff output in the supplementary materials for the reviewers.

We apologize for the delay in submitting this response and greatly appreciate the reviewers' thoughtful feedback. We believe these revisions have significantly strengthened the manuscript and look forward to further discussions and the decision of the reviewers.

Thank you all again,
Authors

---

### Decision · Action_Editor_8x2s · 2025-04-10

**Recommendation:** Accept as is

**Comment:**

In a nutshell, the manuscript has introduced a reasonable method for topology optimization by leveraging Neural Implicit Fields. While the technical novelty has not been highly pronounced,  the engineering pipeline turns out to be useful for a number of tasks related to the increasingly popular field of physics-based machine learning.

**Audience:**

In these years, ML + physics has been a new research theme in the AI community. This will help ground the AI technologies to a larger scope such as AI4Science.

**Claims And Evidence:**

Based on the reviews and the revised version, all the claims in the submission are well grounded.